# The AutoICE Challenge

Andreas Stokholm[2, 6], Jørgen Buus-Hinkler[1], Tore Wulf[1], Anton Korosov[3], Roberto Saldo[2], Leif Toudal Pedersen[2], David Arthurs[4], Ionut Dragan[11], Iacopo Modica[12], Juan Pedro[13], Annekatrien Debien[11], Xinwei Chen[7], Muhammed Patel[7], Fernando Jose Pena Cantu[7], Javier Noa Turnes[7], Jinman Park[7], Linlin Xu[7], Andrea Katharine Scott[8], David Anthony Clausi[7], Yuan Fang[7], Mingzhe Jiang[7], Saeid Taleghanidoozdoozan[7], Neil Curtis Brubacher[7], Armina Soleymani[7], Zacharie Gousseau[7], Michał Smaczny[9], Patryk Kowalski[9], Jacek Komorowski[9], David Rijlaarsdam[10], Jan Nicolaas van Rijn[14], Jens Jakobsen[1], Martin Samuel James Rogers[15], Nick Hughes[16], Tom Zagon[17], Rune Solberg[5], Nicolas Longépé[6], and Matilde Brandt Kreiner[1]

[1]Danish Meteorological Institute (DMI), Copenhagen Denmark
[2]DTU Space, Department of Space Research and Technology, Technical University of Denmark (DTU), Kgs. Lyngby, Denmark
[3]Nansen Environmental and Remote Sensing Center (NERSC), Bergen, Norway
[4]Polar View, Herlev, Denmark
[5]Norwegian Computing Center (NR), Oslo, Norway
[6]ϕ-lab, European Space Research Institute (ESRIN), European Space Agency (ESA), Frascati, Italy
[7]Department of System Design Engineering, University of Waterloo, Waterloo, Canada
[8]Department of Mechanical and Mechatronics Engineering, University of Waterloo, Waterloo, Canada
[9]Warsaw University of Technology, Warsaw, Poland
[10]Ubotica Technologies, Dublin, Ireland
[11]SpaceTec Partners, Brussels, Belgium
[12]GMATICS, Rome, Italy
[13]EarthPulse, Barcelona, Spain
[14]Leiden Institute of Advanced Computer Science, Leiden University, Leiden, The Netherlands
[15]AI Lab, British Antarctic Survey, Cambridge, United Kingdom
[16]Norwegian Ice Service, Norwegian Meteorological Institute, Oslo, Norway
[17]Canadian Ice Service, Environment and Climate Change Canada, Ottawa, Canada

**Correspondence:** Andreas Stokholm (stokholm@space.dtu.dk)

**Abstract.** Mapping sea ice in the Arctic is essential for maritime navigation, and growing vessel traffic highlights the necessity of timeliness and accuracy of sea ice charts. In addition, with the increased availability of satellite imagery, automation is becoming more important. The AutoICE Challenge investigates the possibility of creating deep learning models capable of mapping multi-sea ice parameters automatically from spaceborne Synthetic Aperture Radar (SAR) imagery and assesses the current state of the automatic sea ice mapping scientific field. This was achieved by providing the tools and encouraging participants to adopt the multi-sea ice parameter retrieval paradigm rather than the current focus on single sea ice parameters, such as concentration. The paper documents the efforts, analyses, compares and discusses the performance of the top five participants' submissions. Participants were tasked with the development of machine learning algorithms mapping the total sea ice concentration, stage of development and floe size using a state-of-the-art sea ice dataset with dual-polarised Sentinel-1 SAR images and 22 other relevant variables while using professionally labelled sea ice charts from multiple national ice services as

reference data. The challenge had 129 teams representing a total of 179 participants, with 34 teams delivering 494 submissions, resulting in a participation rate of 26.4%, and was won by a team from the University of Waterloo. Participants were successful in training models capable of retrieving multiple sea ice parameters with convolutional neural network and vision transformer models. The top participants scored best on the total sea ice concentration and stage of development, while the floe size was more difficult. Furthermore, participants offered intriguing approaches and ideas that could help propel future research within automatic sea ice mapping, such as applying high downsampling of SAR data to improve model efficiency and produce better results.

## 1 Introduction

Effective navigation in the cold and remote polar regions requires timely and high-resolution sea ice charts detailing contemporary local ice conditions to circumnavigate or traverse safely and quickly. Therefore, sea ice charts are an indispensable information infrastructure ensuring the transportation of goods and people and supporting activities such as tourism and fishing. The diminishing Arctic sea ice (Perovich et al., 2020) enables new activities, such as shipping avenues using the Northern trade routes or resource prospecting. The Arctic could offer quicker connections between the Atlantic and Pacific oceans with the potential for time and cost savings (Bekkers et al., 2017). Research indicates that ice conditions will become increasingly dynamic, and therefore, it is continuously vital to monitor maritime activities (Boutin et al., 2020). Another use-case for high-resolution ice information is assimilation into weather and climate models for improved performance, as sea ice acts as an intermediate medium between the ocean and the atmosphere, reducing interaction. These models often rely on coarse-resolution sea ice products, e.g. OSI SAF (OSI SAF, 2017) produced by EUMETSAT and based on passive microwave radiometry, and could thus benefit from the higher spatial resolution offered by the SAR-based sea ice maps.

### 1.1 Context

Arctic sea ice is charted by professional sea ice analysts at national ice services worldwide, such as the Greenland Ice Service at the Danish Meteorological Institute (DMI) and the Canadian Ice Service. The charting process follows the SIGRID-3 standard developed by the International Ice Charting Working Group (IICWG) for the World Meteorological Organisation (IICWG, 2010). Over the years, the origin of input data has ranged from airborne campaigns to satellite measurements with multitudes of instruments. The vastness and remoteness of the Arctic pose monitoring challenges that have made satellite observations the universal approach, offering wide coverage, cost savings and high update frequency compared to other monitoring options, such as airborne campaigns. However, optical imagery is unreliable for sea ice monitoring due to a dependency on sunlight (absent during the Arctic winter) and cloud cover, which can be indistinguishable from sea ice. Despite these challenges, when available, optical imagery is still used in operational sea ice charting. The Advanced Microwave Scanning Radiometer 2 (AMSR2) instrument onboard the JAXA GCOM-W1 offer brightness temperature measurements with daily coverage of the Arctic at a resolution in the order of 35x62 km to 3x5 km per pixel (frequency dependent, 6.925 - 89 GHz) (Kasahara et al., 2012), which is insufficient for use in tactical navigation. Instead, active microwave systems like Synthetic Aperture Radar

(SAR) measurements are the backbone for sea ice charting with occasional supplements from other instruments ((Saldo et al., 2021), manual). SAR offer particular versatile measurements in finer than 100m pixel spacing independent of sun illumination and cloud cover. One challenge with SAR data is interpretability, as the radar backscatter depends on surface properties, including roughness, and different surfaces can appear similar. Furthermore, open water and sea ice can resemble one another in their electromagnetic texture appearance (Jackson and Apel, 2004). To provide accurate ice charts, professional ice analysts manually interpret and draw charts based on their in-depth experience and knowledge using Geographical Information System (GIS) software. However, this manual analysis is resource- and time-consuming, constraining the number of daily charts and coverage to the manpower commitment. Naturally, this motivates the development of fully or partially automatic tools that can provide more detailed ice and consistent information for a wider area, delivered near-real-time.

## 1.2 Other relevant works

The interest in automating the retrieval of sea ice information from SAR imagery has been present for decades with early contributions including the usage of texture features as input to support vector machines and other early neural network types (Zakhvatkina et al., 2017; Karvonen, 2014, 2004). Contemporary attempts highlight deep learning, particularly semantic image segmentation, with Convolutional Neural Networks (CNNs) as a primary contender to provide a reliable and precise automatic alternative. An initial study was published by Wang et al. (2016) with additional entries Wang et al. (2017a, b) and continued by Cooke and Scott (2019) highlighting the validity of the approach to map the total Sea Ice Concentration (SIC) in Canada. However, in these early studies, network complexity, data quantity and coverage can be considered limiting factors.

In 2020, an initial version of an open-source deep learning dataset was launched, Automated Sea Ice Product (ASIP) Dataset (ASID-v1) (Malmgren-Hansen et al., 2020). In connection, initial model results using the dataset were published in Malmgren-Hansen et al. (2021) using a custom-built CNN architecture and highlighted early work on the data fusion of SAR and AMSR2 to map the SIC using a regression-based optimisation approach. These models resulted from the first attempts to apply large datasets of multiple 100 GBs for training and emphasised obstacles that became foundations for further model development. E.g. in Heidler et al. (2021), the authors emphasised the importance of a larger receptive field to improve the model's performance developed in Malmgren-Hansen et al. (2021).

The European Space Agency's (ESA) project AI4Arctic continued the efforts of the ASIP project. It produced the second version of the dataset, ASID-v2, in 2021 (Saldo et al., 2021), which became a part of the ESA AI Ready Earth Observation (AIREO) datasets and led to new CNN-related works such as Tamber et al. (2022), and the AI4SeaIce article series (Stokholm et al., 2022; Kucik and Stokholm, 2022, 2023; Stokholm et al., 2023) that has investigated multiple facets of mapping the SIC and approaches to representing it in an optimisation setting. A new study in Wulf et al. (2024) explores data fusion of SAR and passive microwave radiometry to map SIC on a pan-Arctic scale. Other notable sea ice mapping literature entries include Radhakrishnan et al. (2021) utilising curriculum learning and de Gelis et al. (2021) applying the U-Net architecture and underlined obstacles associated with ambiguous SAR signatures and the interest of large receptive fields. In parallel, other efforts include the ExtremeEarth project (Koubarakis et al., 2021) with its polar use-case such as Khaleghian et al. (2021b) focusing on the sea ice type. In addition, there have been several other attempts at mapping sea ice type in SAR imagery using

deep learning-based approaches such as Boulze et al. (2020); Khaleghian et al. (2021b); Liu et al. (2021); Lyu et al. (2022); Jiang et al. (2022); Kortum et al. (2022); Guo et al. (2023). Other literature entries have attempted sea ice floe mapping, such as Chen et al. (2020); Nagi et al. (2021) and discriminating between open water and sea ice in individual pixels (Khaleghian et al., 2021a; Wang et al., 2023; Rogers et al., 2024). Common for these past entries is the focus on single sea ice parameter retrieval. The winners of the AutoICE Challenge document their models' capabilities in multi-sea ice retrieval and perform an ablation study on the model input parameters in Chen et al. (2024a) using the AI4Arctic Challenge Dataset. Furthermore, the authors in Chen et al. (2024b) have shown that detailed SIC maps can be obtained by training models with sea ice charts. Many challenges and advancements within the broader scope of Earth observation and artificial intelligence are highlighted in Tuia et al. (2023).

## 1.3 Objective of the AutoICE Challenge

The objective of the AutoICE challenge was to advance state-of-the-art sea ice parameter retrieval from SAR data with an increased capacity to derive more robust and accurate automated sea ice maps and show that models can retrieve multiple sea ice parameters. In parallel, this provides an opportunity to assess the current state of the scientific field. Furthermore, the challenge has provided a common reference dataset that can be used as a benchmark for comparisons of future model developments.

The field of automatic sea ice mapping has been hastily improving over the past years. However, common for many past literature entries listed here is the focus on single sea ice parameters, either the SIC or type and the regional focus on individual ice services, i.e. Canadian, Greenlandic or Norwegian. Sea ice charts are a treasure trove of expert-labelled training data extending for multiple years and covering vast areas. To propel the automatic sea ice mapping research field towards retrieving multiple sea ice parameters with data from a wider regional area and across national borders - the Artificial Intelligence For Earth Observation (AI4EO) AutoICE Challenge was designed. The challenge aimed to engage and encourage students, sea ice experts, and machine learning practitioners to develop models capable of automatically mapping sea ice and generating new ideas and methods. In addition, the challenge provides a common reference benchmark to support further comparisons in future model developments within the community. Participants were tasked with mapping three sea ice parameters that are all important in describing the composition of the sea ice cover relevant to navigation as well as weather and climate models: SIC, which represents the ratio of sea ice to open water and is the primary descriptor of the sea ice charts. SIC helps ships identify areas of sea ice and the marginal ice zone. The second parameter is the Stage Of Development (SOD), which is the type of sea ice and is a proxy for the age of the ice, which in turn is a proxy for the thickness. The parameter supports decision-making regarding which areas of the ice can be broken by what type of ship. The final ice parameter is the Floe size (FLOE), which characterises the size of ice flakes/floes and aids in determining areas of ice leads and the degree to which the ice is broken into smaller floes. This paper summarises the AutoICE challenge, the AI4Arctic Sea Ice Challenge Dataset, the tools provided to the participants, and the evaluation of submissions. In addition, the results of the top five participants are analysed and compared, and the outcome of the AutoICE Challenge and the state of the automatic sea ice mapping research field are discussed, highlighting avenues for future work.

### 1.4  Article breakdown

Initially, the setup is presented in Section 2, including the evaluation criteria and the tools available to the participants. This is followed by Section 3, describing the challenge data provided by the organisers. Afterwards, in Section 4, an overview of the participation rate is presented together with the final challenge results. 3 of the top 5 teams summarise their solutions in Section 5. This is followed by comparing scene examples from the test dataset in Section 6. Finally, the challenge is discussed and concluded in Sections 7 and 8, highlighting key takeaways and future directions of research to advance the state-of-the-art in automatic sea ice mapping.

## 2  Challenge Setup

An external panel of experts in AI and sea ice charting was appointed to help design and evaluate the challenge. The expert panel members included two sea ice charting experts appointed by the International Ice Charting Working Group (IICWG) and represented universities and research institutes. The expert panel participated in a dedicated workshop hosted by the organizers to discuss submission evaluation metrics and setup, etc.

The challenge was designed to cater to a large audience by providing manageable resources and a clear and purposeful objective. Participants were given a state-of-the-art dataset, the ASID Challenge dataset (Buus-Hinkler et al., 2022a), to train their models. The dataset encompasses remotely sensed data from multiple sensors, geographical information and atmospheric and land-surface quantities from reanalysis models to encourage diverse data fusion methodologies. The dataset spanned multiple years and charts from multiple national ice services (Canada and Greenland). Two versions of the dataset were prepared, an unaltered (raw) version and a Ready-To-Train (RTT) version, to cater to the ease of getting started while simultaneously allowing those who prefer fully customised model training setups to pursue their ideas. The scenes were divided into 513 for training (Buus-Hinkler et al., 2023a, b) and 20 for testing (Buus-Hinkler et al., 2022b, c). Participants could not access the testing scenes' sea ice charts to prevent overfitting to the set during training. The testing scenes contained all ice classes in the training dataset. They were selected to represent various sea ice SAR signatures with charts from the Canadian and Greenland Ice Services from January 2018 to December 2021.

Participants were also provided with get-started tools consisting of software created by the organisers to help get started and training models for the challenge with the RTT dataset. In addition, computing resources through the Polar Thematic Exploitation Platform (PolarTEP) were available to participants. The challenge was hosted on the ESA-funded AI4EO.eu challenge platform, introducing the challenge design and rules, links to the dataset and tools and a submission portal with an associated leaderboard, where participants could compare their results to those of other teams. The competition launched on November 23 2022 and closed on April 17 2023.

## Metrics and final evaluation

To submit a solution, participants produced maps of the three sea ice parameters at an 80m pixel spacing and uploaded them to the AI4EO.eu platform portal. The platform backend computed a score based on comparing the reference data and provided the score to the teams. A public and private score was calculated. The public score was calculated based on 10 of the 20 test scenes and the private score on all 20 scenes. The (team's best) public score was shown on the leaderboard. In contrast, the private score was withheld from the participants until the competition's closure and used as the final ranking of the teams to prevent overfitting to the test dataset.

The participant's test set solutions were evaluated based on a weighted sum of three metrics, one for each of the three sea ice parameters. The SIC score was evaluated using the $R^2$ coefficient. $R^2$ captures the regression aspect of sea ice concentrations (inter-class relationship, i.e. 10% SIC being closer to 20% than 30%) and can be expressed as a percentage. It is formulated as:

$$R^2 = 1 - \frac{\sum_{i=1}^{N_{\text{pixel}}} (y_i^{true} - y_i^{\text{pred}})^2}{\sum_{i=1}^{N_{\text{pixel}}} (y_i^{true} - \hat{y}^{true})^2} \tag{1}$$

where $y_i^{true}$ is the true $i^{th}$ pixel, $\hat{y}^{true}$ is the mean true pixel value, and $y_i^{\text{pred}}$ is the predicted class of the $i^{th}$ pixel.

The SOD and FLOE parameters were both evaluated using the F1 score. SOD and FLOE categories, as opposed to SIC, are not directly linked, and thus, a classification-oriented metric was deemed suitable for this evaluation. F1 is the harmonic mean of each class's precision and recall metrics. The F1 score for each ice parameter considers the dataset sea ice class imbalance by accounting for the number of pixels for each class. The F1 score can further be expressed as a percentage and is formulated as follows:

$$F1 = 2 \frac{precision \cdot recall}{precision + recall} \quad , \text{where} \quad precision = \frac{T_P}{T_P + F_P} \quad \text{and} \quad recall = \frac{T_P}{T_P + F_N} \tag{2}$$

Here, $T_P$ is the number of true positives, $F_P$ is the number of false positives, and $F_N$ is the number of false negatives.

The three sea ice parameter scores were combined into one final score using a weighting scheme. With input from the expert panel, the final score emphasised SIC and SOD over FLOE, as FLOE was deemed less important for the ice service and users by the ice charting experts. The weights were $\frac{2}{5}$ for both SIC and SOD, and $\frac{1}{5}$ for FLOE. For the metric calculations, pixels that did not contain a sea ice class, e.g. land, were discounted.

## Get-started tools

To increase the accessibility of the challenge, get-started tools with Python functions and three notebooks were prepared by the organisers. One notebook served as a thematic and data exploration introduction. Another provided a model training setup implemented in PyTorch and, finally, a notebook to produce a test solution. In addition, a simple U-Net model implemented in PyTorch provided a common starting point for participants. These files and notebooks offer examples of how to carry out model training experiments but were not required to be used.

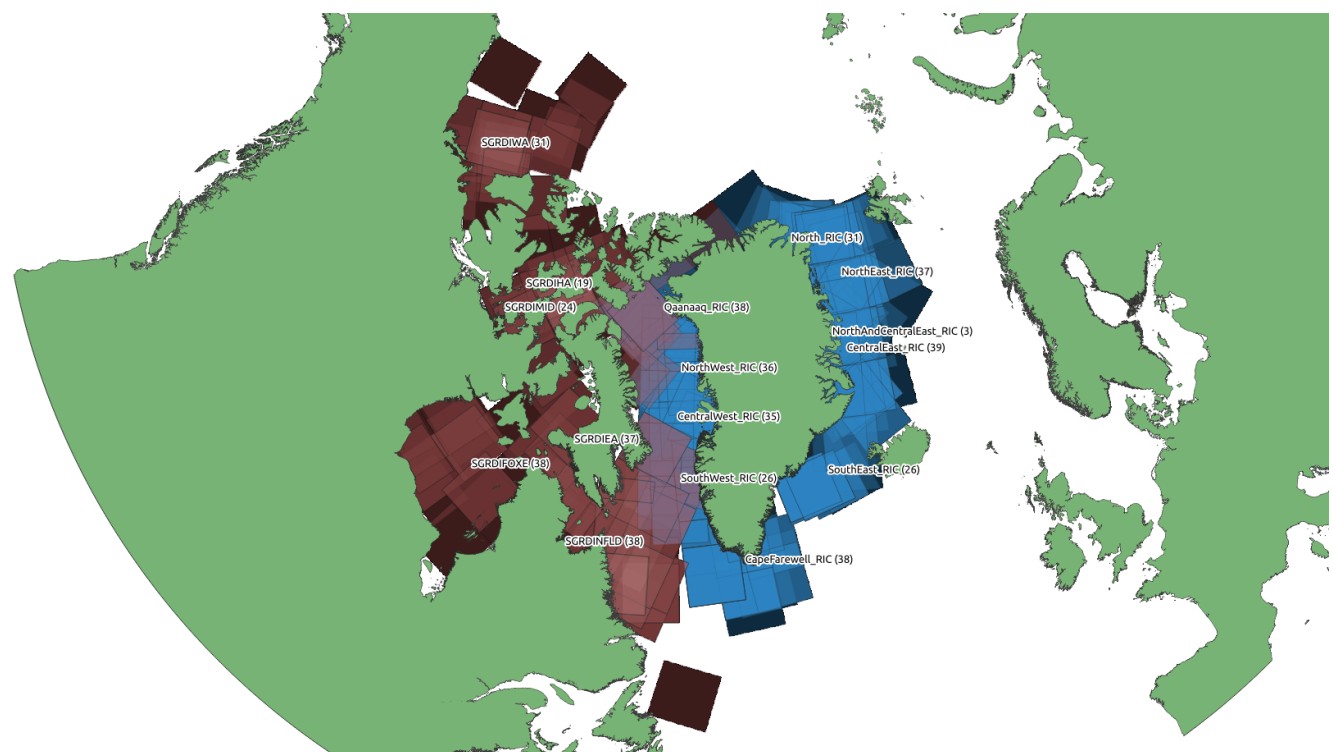

**Figure 1.** Overview of the 513 training scenes in the AI4Arctic Sea Ice Challenge Dataset. Red and blue squares illustrate scenes with ice charts from the Canadian and Greenland ice services, respectively. Increasingly bright colours indicate a larger number of charts.

## 3   The AI4Arctic Sea Ice Challenge Dataset

The AI4Arctic Sea Ice Challenge Dataset (ASID Challenge) includes 533 co-located and georeferenced scenes between January 2018 and December 2021 distributed across the Canadian and Greenlandic Arctic as illustrated in Fig. 1. In this section, the data variables are examined briefly. Please see the official dataset manual in (Buus-Hinkler et al., 2022a) for more details. Each scene contains sea ice chart reference data, SAR images, passive microwave radiometry measurements, and numerical weather prediction parameters. The following subsections describe these data sources and a prepared and ready-to-train dataset.

**Sea ice charts - reference data**

Sea ice charts describe the local ice conditions at the time of acquisition of the input satellite imagery, based on professional interpretations of SAR images and represented distinctly as polygons of relatively homogeneous areas of sea ice, steered by the common guidelines outlined in the SIGRID-3 standard but still subject to individual interpretation. There is a natural limitation to how many details and polygons the ice analysts can manually draw within the chosen scale and coverage of

the ice map. At the same time, there is a focus on safety and not delaying the information to the users more than necessary. Therefore, the polygons' boundaries are very accurately drawn but cover large areas with a subsequent low effective resolution.

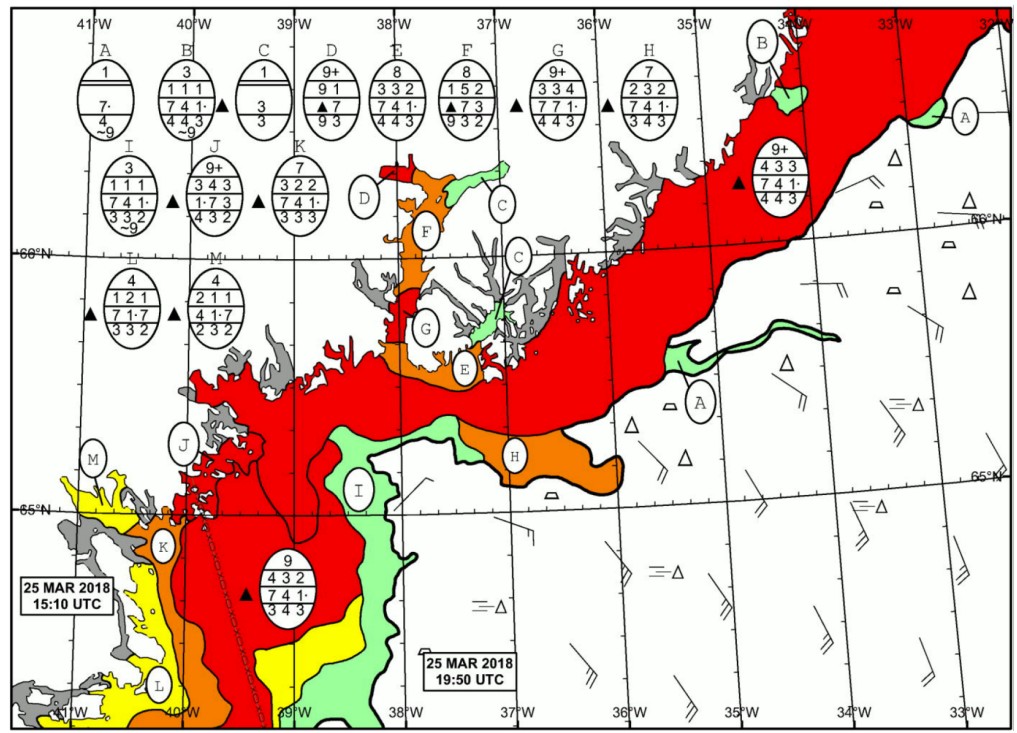

**Figure 2.** Manually produced sea ice chart from the Greenland Ice Service containing polygons with an associated ice "egg code" describing ice conditions. The image is depicted in geographical coordinates. Greenland Strait, Southeast Greenland. The scene was acquired on March 25, 2018.

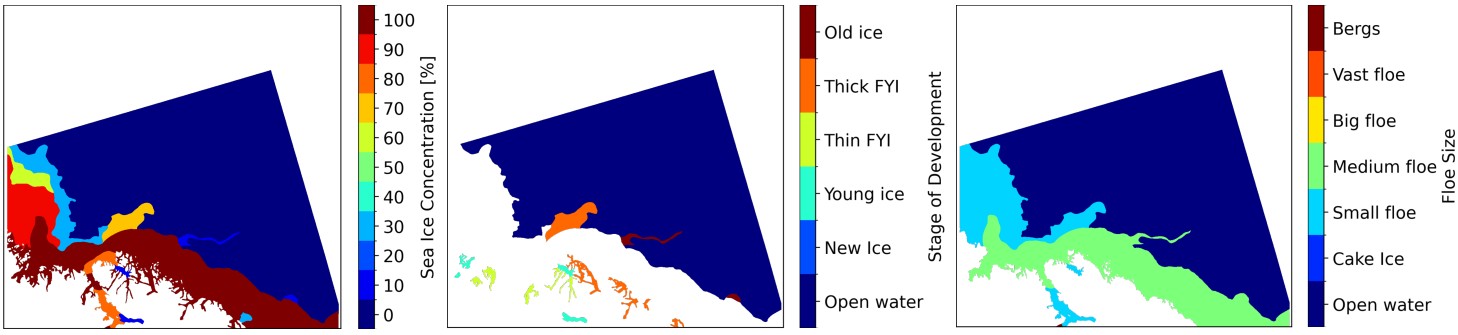

**Figure 3.** SIC, SOD and FLOE maps from the ice chart in Fig. 2. White pixels are masked areas from either no information, land or ambiguous polygons with no dominant ice class for the respective parameter. The colour code is slightly different than Fig. 2. The images are depicted in the original SAR geometry.

Understandably, manual production cannot relay ice information with a level of detail that matches the high-resolution and multidimensional electromagnetic SAR textures.

Studies have suggested that the SIC between ice analysts can vary on average 20% and, in worst cases, up to 60% (Karvonen et al., 2015). Similarly, low SICs (10-30%) can be overestimated while middle SIC classes (50-60%) can exhibit a wide spread with high variability Cheng et al. (2020). Additionally, the marginal ice zone typically receives more attention during the analysis, as these areas see higher maritime activity ((Saldo et al., 2021), manual). Despite these uncertainties, pixels in the sea ice charts are treated as equally valid.

The sea ice charts used in the challenge dataset are either produced by the Canadian Ice Service (CIS) or the Greenland Ice Service at DMI, illustrated in Fig. 1 in red and blue, respectively, with a lighter colour indicating more scenes. Each chart is temporally and geographically matched with a Sentinel-1 image within 5 or 15 minutes of the timestamp of the DMI and CIS ice charts. The original ice chart data is contained in an ESRI ShapeFile format, projected to the Sentinel-1 SAR geometry and rasterized to a map matching the pixel spacing of the SAR image with polygon IDs and an associated ice information look-up table. In the RTT dataset version, the ice chart was converted into three maps, one for SIC, SOD and FLOE, using the ice codes defined in the SIGRID-3 convention. SIC is converted into 11 classes from 0-100% in discrete increments of 10%, the SOD into 6 classes; open water, *new ice*, *young ice*, *thin First-Year Ice* (FYI), *thick FYI*, and *old ice*. FLOE is converted into 7 classes; open water, *cake ice*, *small*, *medium*, *big*, *vast* floes as well as *bergs*. Some of these classes result from merging multiple approximate ice codes, as highlighted in the dataset manual (Buus-Hinkler et al., 2022a). In addition, as the SOD and FLOE are given as partial SOD or FLOE concentrations, there may be multiple categories of SOD or FLOE mixed within each ice polygon without the exact location provided. To select the SOD or FLOE class while minimising ambiguity, the SOD or FLOE class must be dominant. Here, we defined a SOD or FLOE class as dominant if the associated partial concentration is at least 65%. Therefore, there are numerous polygons where a total SIC exists, but the polygon does not have an associated SOD and/or FLOE. Despite this effort, multiple classes may still be mixed in each polygon. An ice chart conversion Python script was provided for participants wishing to use the raw dataset. The three sea ice parameter maps associated with Fig. 2 are illustrated in Fig. 3, shown in the original SAR measurement geometry. In this example, there are polygons without a dominant SOD and FLOE, which are shown as white - similar to land or areas with no measurement values.

**Synthetic Aperture Radar**

The primary data source is the two-channel dual polarized (HH and HV) Sentinel-1 C-band 5.410 GHz frequency level 1 Ground Range Detected Medium resolution images acquired in the Extra-Wide operational mode (Torres et al., 2012). The SAR image has been noise corrected using the algorithm described in Korosov et al. (2022). In addition, the SAR incidence angles and a pixel-wise distance-to-land map are included. The closest temporally overlapping SAR image to the ice chart in Fig. 2 and ice parameters in Fig. 3 are illustrated in Fig. 4 in the original SAR geometry.

**Passive microwave radiometry**

The challenge dataset also contains overlapping level-1b brightness temperatures measured with the AMSR2 passive microwave radiometer onboard the JAXA GCOM-W satellite. The maximum time difference between the acquisition time of the Sentinel-1 image and the overlapping AMSR2 swath is 7 hours. The AMSR2 measurements are resampled to the Sentinel-1 ge-

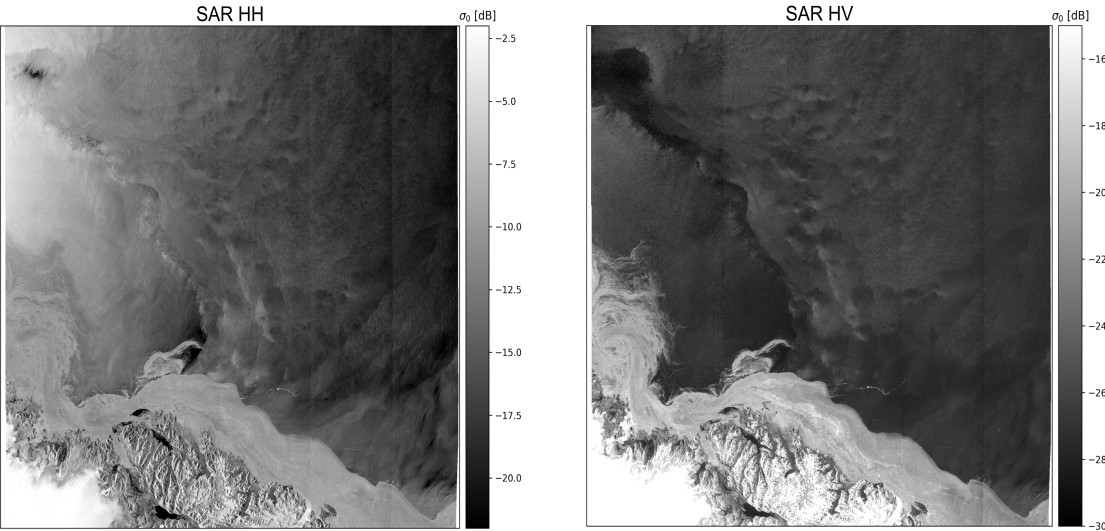

**Figure 4.** HH and HV SAR images corresponding to the ice chart illustrated in Fig. 2 and 3 in $\sigma_0$ dB backscatter values and depicted in the SAR acquisition geometry.

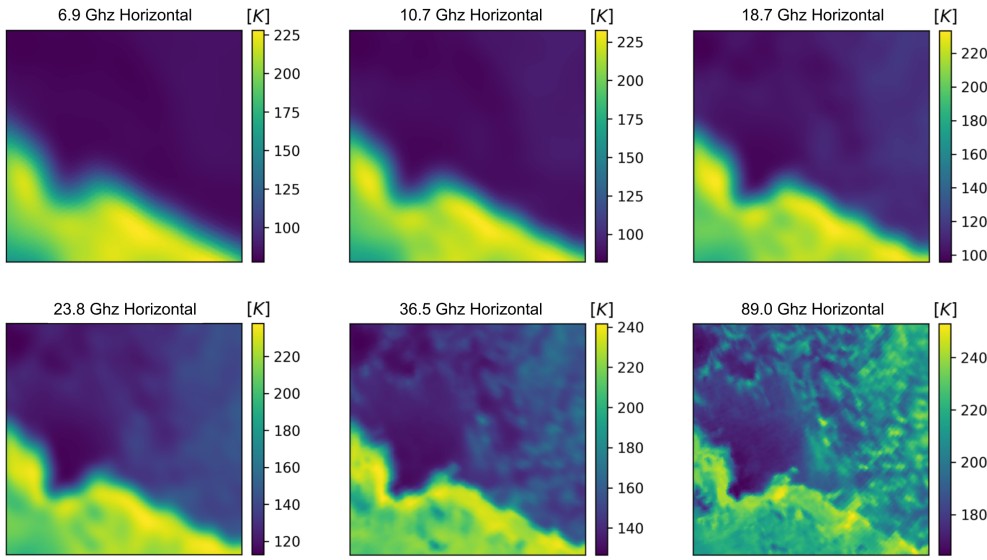

**Figure 5.** An example of the available horizontally polarised brightness temperatures in Kelvin from the AMSR2 passive microwave radiometer onboard the JAXA GCOM-W Satellite covering the scene in Fig. 2 and viewed in the same perspective as Fig. 3 and 4.

ometry to the coordinates of every 50 by 50 (2 km) pixel using a gaussian weighted interpolation for each polarization (vertical and horizontal) and frequency (6.9, 7.3, 10.7, 18.7, 23.8, 36.5, 89.0 GHz). Examples of AMSR2 measurements corresponding to the ice maps in Fig. 3 and the SAR data in Fig. 4 are illustrated in Fig. 5. Auxiliary AMSR2 variables include the AMSR2

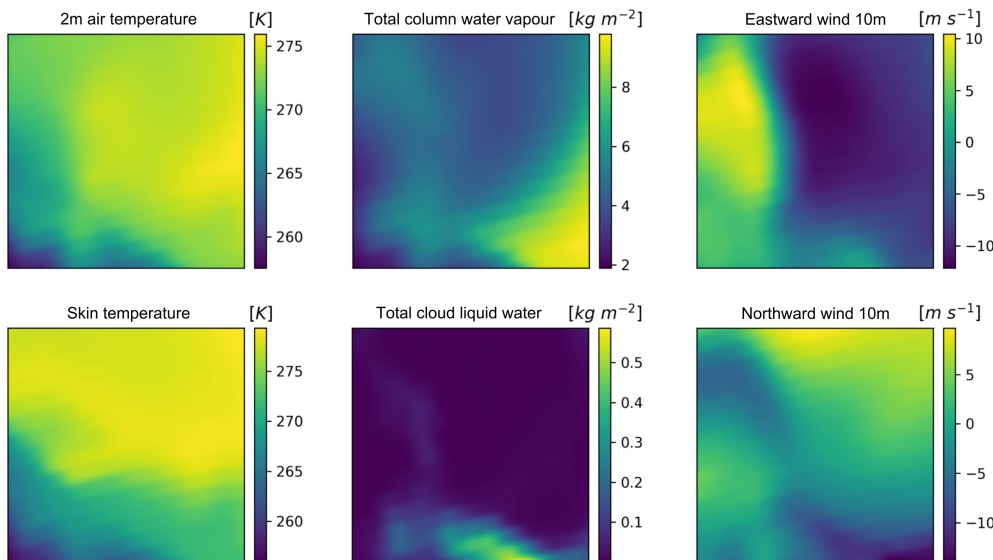

**Figure 6.** An example of the numerical weather prediction parameters including 2m air and skin temperature, total column water vapour and cloud liquid water, and East- and Northward wind 10m components for the scene depicted in Fig. 2 and in the same perspective as Fig. 3-5.

swath names of the used AMSR2 level-1b product(s), AMSR2 swath numbers (relevant when mosaicing multiple swaths), and the time delay(s) between AMSR2 and Sentinel-1.

**Numerical weather prediction parameters**

Several numerical weather prediction parameters from the ERA5 (ECMWF Reanalysis v5) are included (Hersbach et al., 2023). The parameters are resampled to the Sentinel-1 geometry in the same manner as the AMSR2 brightness temperatures using a

gaussian weighted interpolation. The parameters are illustrated in Fig. 6 and encompass the 2-meter air and skin temperature, the total column water vapour and cloud liquid water, and the eastward and northward 10-meter wind components rotated to account for the Sentinel-1 flight direction.

**Ready-To-Train (RTT) dataset version**

Some preprocessing choices were already made for the participants for the RTT dataset version. To reduce the barrier of entry,

the original 40m pixel spacing (∼10,000 x 10,000 pixels) in the SAR (and ice charts, etc.) data was downsampled to 80m (∼5,000 x 5,000 pixels). The participants were also required to deliver sea ice maps in this pixel spacing. The SAR image, distance-to-land map and incidence angle data were downsampled using a 2×2 averaging kernel, whereas ice charts were reduced spatially using a 2x2 max kernel. This is followed by an alignment of masks (nan-values) across the data, except the sub-gridded variables, e.g. AMSR2 data, and the SOD and FLOE polygons with no dominant ice code. Afterwards, the

scenes were standard-scaled using the mean and standard deviation of all training data within each data channel. Finally, pixels

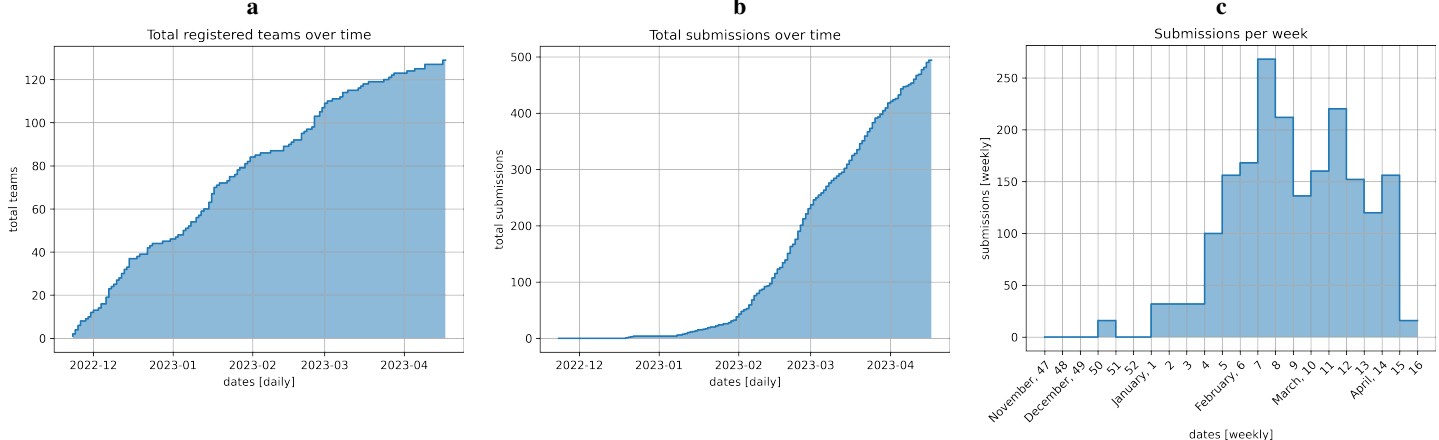

**Figure 7.** User challenge engagement. **a**): shows the total accumulated registered teams per day on the AI4EO platform. **b**): illustrates the total accumulated submissions per day during the competition. **c**): highlights the number of submissions per week over the course of the challenge.

without ice chart values were replaced with the values 2 and 255 in the SAR images and ice charts, respectively. This was carried out to represent non-data or masked pixels and enable the discounting of these pixels during loss optimisation and the computation of the evaluation metrics.

## 4 Participation and submission results

The competition received good traction from a diverse set of stakeholders ranging from academics, students and industry. Fig. 7 illustrates (**a**) the total number of registered teams (multiple users can be within each team) over the course of the competition, (**b**) the total number of team submissions, and (**c**) the total number of submissions per week. The competition saw a continuous influx of users with fewer registrations towards the end of the competition. At the end of the competition, a total of 129 teams and 179 associated users had registered. Participants started delivering their test set solutions around halfway through the competition, with a peaking submission rate of around two-thirds of the way through the competition and a spike in submissions nearing the closing date. In total, 494 test solutions were submitted from 34 different teams.

### 4.1 Submission results

The top-performing teams are listed in Tab. 1, showcasing the combined final private score, as well as the private score for each ice parameter and the number of submissions per team. In the bottom row, the mean and standard deviation (STD) of the top 5 teams are included. The overall winner of the AutoICE challenge was the combined team from the Department of System Design Engineering at the University of Waterloo (*UW*), encompassing a total of 14 people. *UW* achieved a combined final score of 86.39%. In addition, the team scored highest on both the SIC and SOD while scoring the lowest among the top 5

teams on the FLOE. The *UW* team consisted of PostDocs and PhD and Master students, were supervised by faculty staff and were very engaged during the competition. In total, *UW* submitted test solutions from a total of 7 team accounts that were all placed in the top 7 on the leaderboard. In total, *UW* submitted 170 test solutions across their team accounts, which was more than the other 4 top 5 teams combined.

The second place went to the team *PWGSN*, two computer science master students and their PhD candidate supervisor from the Warsaw University of Technology with a combined score of 82.48%, the highest score on FLOE and a total of 42 submissions. In third place, the team *crissy* scored 81.17% with a single submission. Fourth went to *sim*, an engineer at Ubotica Technologies who submitted 7 test solutions with a score of 80.61%. Finally, on the fifth, *jff* scored 80.56% with a total of 59 submissions. *crissy* and *jff* has not shared their affiliations.

From the top 5 participants' mean and STD ice parameter scores in Tab. 1, it is clear that the SIC was the variable that all participants scored the highest numerical percentage on, followed by the SOD and finally FLOE. The STD appear to be highest for the SOD, though skewed by the high *UW* performance. Excluding the *UW* SOD score, the SOD STD would be the lowest among the three ice parameters at 1.5, compared to 1.8 and 2.4 for SIC and FLOE, respectively.

**Table 1.** Final ranking and scoring of the top 5 participating teams including the individual ice parameter scores, the mean ice parameter scores across the teams with the standard deviation, and the total submissions for each team.

| Rank | Team | Final | SIC | SOD | FLOE | Submissions |
|---|---|---|---|---|---|---|
| 1 | **University of Waterloo** | **86.39%** | **92.02%** | **88.61%** | 70.70% | 170 |
| 2 | PWGSN | 82.48% | 89.70% | 76.94% | **79.12%** | 42 |
| 3 | crissy | 81.17% | 85.35% | 80.26% | 74.66% | 1 |
| 4 | sim | 80.61% | 87.22% | 77.52% | 73.59% | 7 |
| 5 | jff | 80.56% | 86.68% | 77.18% | 75.10% | 59 |
| mean | | - | 88.19% ±2.66 | 80.10% ±4.94 | 74.63% ±3.04 | - |

## 5 Top submission solutions

In the following subsections, three of the top 5 teams - UW, PWGSN and sim teams - have contributed short descriptions of their model solutions. The two remaining teams have not provided information about their personal models. All participants in the AutoICE challenge were invited to submit a full description of their solutions to the special issue in the Cryosphere. "*AutoICE: results of the sea ice classification challenge*". In the proceeding, *we* refer to the individual teams describing their solutions.

### 5.1 Rank 1 - University of Waterloo

To streamline our model development process, we utilized the RTT version of the AI4Arctic Sea Ice Challenge Dataset (Buus-Hinkler et al., 2023b). To ensure consistent predictions with the ice chart-derived label maps, it is crucial to increase the

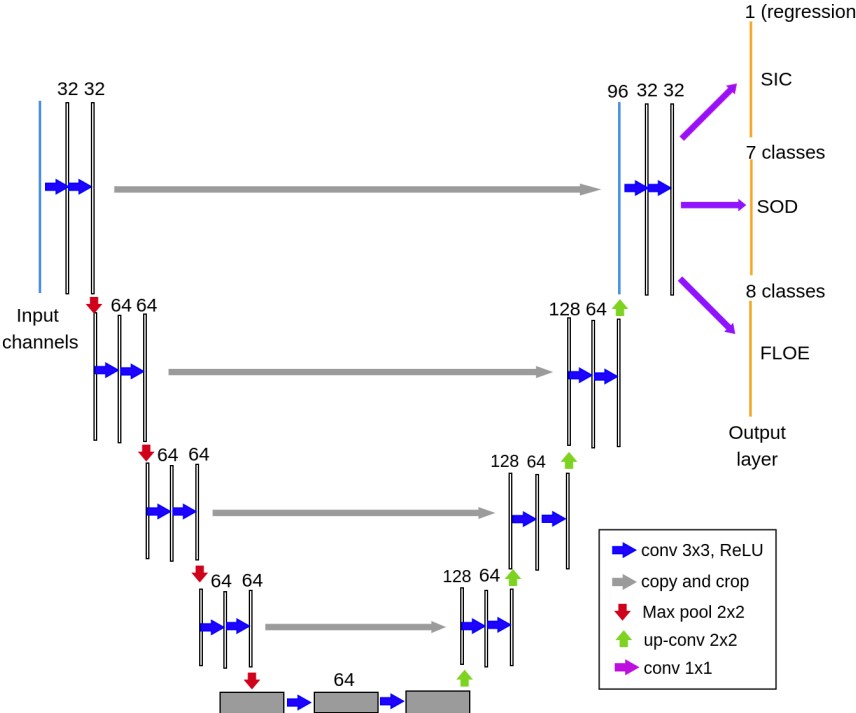

**Figure 8.** The structure of the multitask U-Net-based model with output layers in yellow utilised by the *UW*-team.

geographical field of view of the model. To achieve this, we downsample the dual-polarized SAR images, distance maps, and corresponding ice chart-derived label maps by a specific ratio (10 in this work). We randomly extract patches of size $256 \times 256$ during training from the downsampled SAR images. The AMSR2 and ERA5 variables are also resampled to the same size and interpolated within the geographical areas covered by the patches. For validation and testing, entire SAR scenes and distance maps are downscaled and combined with other upsampled variables as input to the trained model. The outputs are then interpolated back to the original size for evaluation. Tab. 2 lists the best combination of input variables. Additionally, to incorporate spatial and temporal information, we interpolate the latitude and longitude coordinates of the Sentinel-1 SAR geographic grid points to match the size of the input SAR image and add the acquisition month of each SAR scene to each pixel to represent the time information.

Regarding the model architecture, we construct a multi-task U-Net that simultaneously estimates three sea ice parameters, as depicted in Fig. 8. It consists of four encoder-decoder blocks with varying numbers of filters. To generate predictions for SOD and FLOE, the output feature maps from the final decoder are separately fed into $1 \times 1$ convolution layers with the number of filters corresponding to the number of classes. This generates pixel-based classification results (i.e., segmentation). As for SIC, a regression head is added at the end to produce SIC estimates. The model training details, including hyperparameter combinations that yield the best validation accuracy, are specified in Tab. 3. We employ Cosine Annealing as a learning rate schedule, which allows the model to converge to a reasonable solution by cyclically adjusting the learning rate. Each epoch

**Table 2.** The combination of input variables that produced the highest score for the *UW*-team.

| Feature abbreviation | Variable description | Total number of channels |
|---|---|---|
| HH, HV, IA | Dual-pol SAR scene with incidence angle information | 3 |
| DM | Distance-to-land map for all pixels | 1 |
| AMSR2 subset | Dual-pol AMSR2 brightness temperature data in 18.7 and 36.5 GHz | 4 |
| ERA5 subset | 10-m wind speed, 2-m air temperature, total column water vapour, total column cloud liquid water | 5 |
| Loc, time | Latitude/longitude of each pixel and scene acquisition month | 3 |

comprises 500 iterations, with patches randomly sampled from the training scenes in each iteration. Through experimentation, we determine that using mean square error (MSE) loss for SIC and Cross-Entropy (CE) loss for SOD and FLOE achieves the highest testing accuracy. To expedite the convergence of the three scores, we assign a larger weight value to the CE losses relative to the MSE loss, as shown in Tab. 3. To ensure consistency between validation and testing accuracy, we select 18 SAR
scenes from the training data that closely match the acquisition locations and time periods of the testing scenes, creating a separate validation set. A combined score, following the metrics given in the competition, is calculated from the validation set at the end of each epoch. The model parameters are updated and saved if the current epoch's score surpasses all previous scores. The final saved model is employed to generate predictions for the testing data. All experiments are conducted on the Narval cluster of Compute Canada (Baldwin, 2012) using an NVIDIA A100-SXM4-40GB GPU and 128GB of memory with
the PyTorch 1.12 library.

**Table 3.** Training specifications of the *UW*-team model solution.

| Optimizer | SGDM |
|---|---|
| Learning rate | 0.001 |
| Weight decay | 0.01 |
| Scheduler | Cosine Annealing |
| Batch size | 16 |
| Number of iterations per epoch | 500 |
| Total epoch | 300 |
| Number of epochs for the first restart | 20 |
| Downscaling ratio | 10 |
| Data augmentation | Rotation, flip, random scale, cutmix |
| Patch size | 256 |
| Loss functions | MSE for SIC, CE for SOD and FLOE |
| Total loss calculation | SIC×1+SOD×3+FLOE×3 |
| Number of validation scenes | 18 |

Among the submissions from over 30 teams worldwide, our method achieved the highest combined score of approximately 86.4%. In particular, it was observed that our method outperformed other methods on SOD (8 percentage points higher than the following best) and SIC scores (2 percentage points higher than the next best). As the ice chart-derived labels for the testing data were released after the competition ended, a comprehensive analysis of the experimental results will be presented in a forthcoming paper for publication.

## 5.2 Rank 2 - PWGSN

For all experiments conducted during the competition, the RTT version of the AI4Arctic Sea Ice Challenge Dataset (Buus-Hinkler et al., 2023b) was used. The data was split into training (502 scenes) and validation (10 scenes) datasets. An epoch was defined as an iteration over all available training scenes. During each step, one patch size of $224 \times 224$ pixels was selected according to the undersampling procedure in Fig. 9.

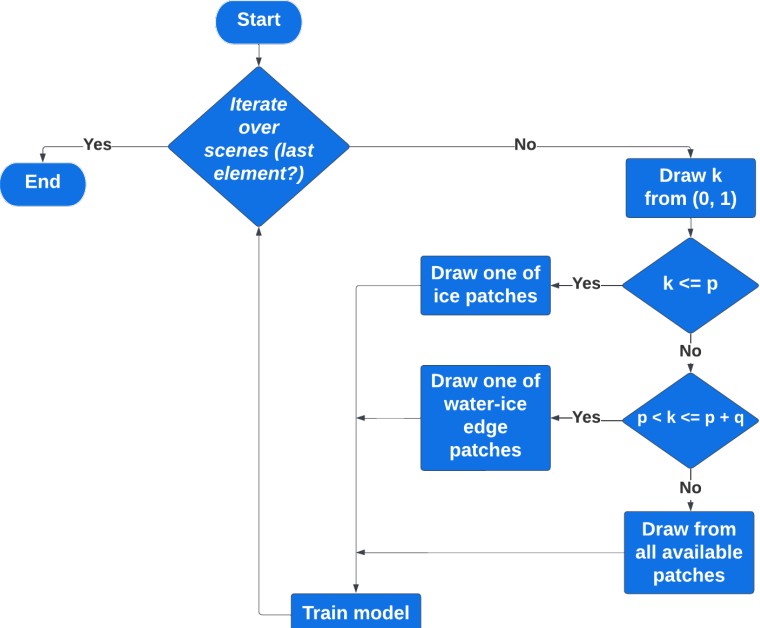

**Figure 9.** The undersampling procedure used by the *PWGSN*-team. Composing training dataset with approximately imposed class frequency.

Each scene was divided into a grid of patches using a sliding window of size $224 \times 224$ pixels with a step of 22 pixels. Each patch was classified into one of the three possible classes, depending on the share of open water pixels (s): open water ($s \geq 0.9$), water-ice edge ($0.2 \leq s < 0.9$) and ice ($s < 0.2$).

During an epoch, patches were randomly selected from scenes to approximately satisfy the predefined class distribution, implying that the share of ice-only patches in an epoch should be close to some value, $p$, and the share of water-ice edge patches should be close to another value, $q$, but without taking image, region or season into account. We have achieved the best results for p = 0.1 and q = 0.2. Training examples were collected into 8-element batches. Training observations consisted

of all 24 channels available in the data. Low-resolution channels were upsampled to the size of SAR images. The following data augmentations were applied in an effort to mitigate overfitting: rotations, flips, multiplicative noise and slight distortions. It is worth noting that only transformer-based architectures were prone to overfitting - for other architectures tested, including CNNs, data augmentations had no positive impact on the metrics.

A modified semantic segmentation model was used with an adjusted number of output heads. This approach enabled us to make SIC, SOD and FLOE maps simultaneously. The model returned three 3-dimensional tensors with an estimated likelihood of pixels belonging to a particular class. An ensemble of 10 models was created to generate the final results. Outputs from each of the models were merged using a majority voting mechanism. All of the models in the ensemble shared the same architecture but differed in the saved model parameter checkpoint, loss function, augmentations and the imposed data distribution.

Validation and test scenes were divided into patches ($224 \times 224$ pixels, $512 \times 512$ pixels and $1024 \times 1024$ pixels) due to memory limitations on the utilised GPU. Predictions were made on each patch separately and then combined together into the final outcome. Different tiling techniques were used during this process, including overlapping the patches, rotating and averaging, and smooth blending, as inspired by Chevalier (2017).

Several convolutional and vision transformer architectures, including EfficientUNet, ResNeXt and DeepLabV3, have been tested. The most promising results were achieved with a transformer-based architecture, where an altered Coat-Lite Medium (Xu et al., 2021) was used as the encoder and an altered Daformer (Hoyer et al., 2022) as the decoder. For the encoder part, transfer learning was applied (weights were pre-trained on the ImageNet dataset). The architecture was inspired by a Kaggle competition notebook (Cijov, 2022). The number of the encoder input channels was adjusted to 24 dimensions available in the RTT dataset. For this purpose, the pre-trained weights of the first convolution layer were averaged and then expanded to the required quantity of input channels.

The distribution of classes for all three maps was highly unbalanced. Thus, experiments were set up in which models were trained using Cross-Entropy (CE), Weighted Cross-Entropy (WCE), focal, dice and ordinal loss. The best results were obtained with CE, WCE and a mixture of CE and dice loss with the ratio of $\frac{0.7}{0.3}$, respectively. The Adam optimizer was applied during the training. In most of the experiments, models were trained in two steps. At first, the learning rate was set to $\eta = 10^{-4}$ while training the model for approximately 50 epochs. Then the model was fine-tuned for subsequent 300 epochs with $\eta = 10^{-5}$.

### 5.3 Rank 4 - sim

Our best solution was based on U-net architecture (Ronneberger et al., 2015), which effectively captures spatial information and preserves fine details, making it suitable for tasks requiring pixel-level segmentation accuracy. One of the primary reasons for choosing U-net as our baseline architecture is our team's prior experience with the architecture. The U-net model has been previously adopted for earth observation data processing tasks, particularly for onboard processing, where limited computational resources are available. The architecture performs well on edge processing hardware (Dunkel et al., 2022). In addition, the architecture required minimal adaptation from the provided code base, enabling us to focus on optimizing its performance for the challenge.

In search for an appropriate network architecture, a secondary solution was explored in the form of DeepLabV3 (Chen et al., 2017). The reasoning for choosing DeepLabV3 was that this state-of-the-art deep learning architecture for semantic image segmentation could be used to segment sea ice since it excels in capturing multi-scale contextual information. We hypothesized that this could improve overall performance due to the scale differences of ice formations. However, despite reasonable results, we did not find this model architecture to be able to outperform our U-Net solution. In addition, due to the larger size of the network, iterations proved more time-consuming than the U-Net architecture and more demanding in terms of computational resources.

For our training pipeline, we utilized the code and data provided by the competition organizers as these resources proved a powerful starting point. Both the get-started notebook and the RTT dataset (Buus-Hinkler et al., 2023b) were leveraged.

The U-Net was trained with the Adam optimizer, with a learning rate of $10^{-4}$, and using a CE loss function. 225 epochs with 100 iterations per epoch were used. For each iteration, a batch was filled with 32 random crops of 256x256 pixels. All input channels were used as input, i.e. the input tensor was of shape [32, 24, 256, 256] [batch, channel, H, W]. This model reached an overall score of 80.6%, with a score of 87.2% for the SIC, 77.5% for the SOD, and 73.6% for the FLOE.

Our best DeepLabV3 model was trained for 76 epochs with 500 iterations per epoch and a batch size of 8. For this training pipeline, the Adam optimizer was used with a learning rate of $10^{-5}$ and CE loss as well. Our best-performing DeepLabV3 model performed worse overall than our U-net, with a public score of 79.1%. Interestingly, the network outperformed U-Net significantly on FLOE segmentation with a public score of 74.5%. For SIC the best public score was close to U-Net with 84.8% and SOD was significantly worse with 75.8%. These results suggest that an ensemble of multiple network architectures could potentially outperform a standalone model by leveraging their complementary strengths. However, due to time constraints, further investigations into ensemble models were not pursued.

## 6   Comparison of top 5 submissions

For a deeper dive into the solution results submitted by the top 5 teams, output maps for two example SAR scenes are highlighted. Fig. 10 illustrates a scene in the native measurement geometry from Hudson Bay in the Canadian Arctic, captured in July 2018, along with a sea ice chart from the Canadian Ice Service. In the top row, the Sentinel-1 HH and HV channels are shown, followed by rows for the SIC, SOD and FLOE ice parameters. The columns show the reference sea ice chart, the solutions by the top 5 teams, the STD between the solutions and the accumulated error between each solution and the reference. The STD and error colour scales are from 0 to the maximum STD, defined as the maximal possible STD. Likewise, the error goes from 0 to the maximal possible accumulated error across the solutions. The STD indicates where the top 5 solutions disagree, while the error shows locations where the top 5 solutions disagree with the reference.

The scene in Fig. 10 was acquired during the Arctic Summer season and showcases sea ice in warm conditions with varying SICs covering the majority of the scene while most ice is in the right-hand side of the image with an area of ice in the centre of the image stretching towards the left side. The top 5 solutions agree on the separation between open water and sea ice. The SIC solutions have the lowest STD among the three sea ice parameters, and the errors are most prominent near the ice-

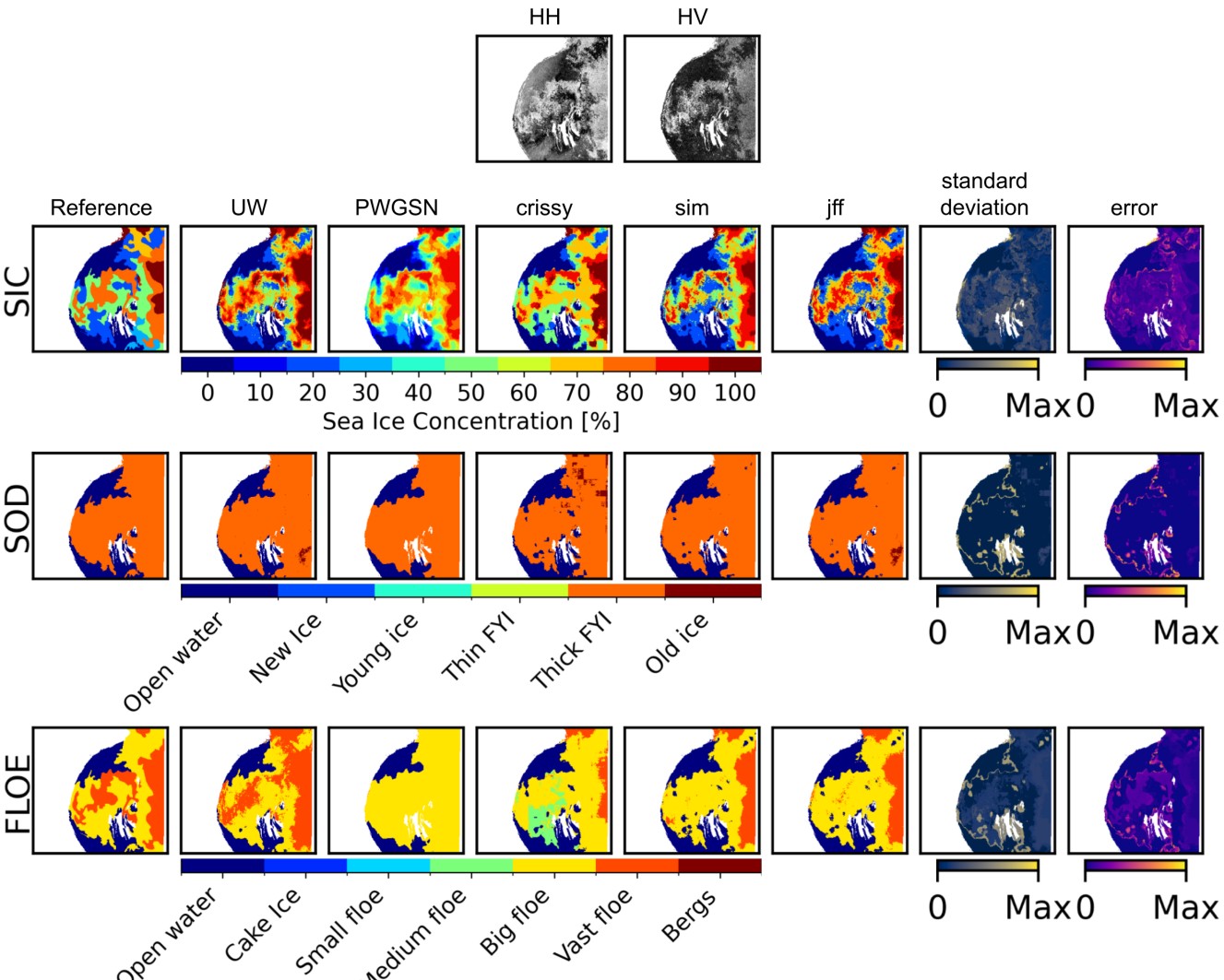

**Figure 10.** Hudson Bay, Canada. First row: SAR HH and HV images, acquired on July 7 2018. Reference ice chart labelled by the Canadian Ice Service. Second row: SIC reference and top 5 solution SIC maps with standard deviation between solutions and accumulated map of error between solutions and the reference. Max indicates the maximum possible standard deviation of 4.9, 2.4 and 2.9 for SIC, SOD and FLOE, respectively, or max accumulated error assuming a linear distance between classes of 50, 25 and 30 for SIC, SOD and FLOE, respectively. The third row contains the SOD and the fourth the FLOE. White areas indicate a mask of either land, with no information or ice polygons without a dominant ice code.

water boundaries and in the upper portion of the scene. In the SOD maps, all solutions are consistent in that they agree on the dominant class being *thick FYI* but disagree on the location of the ice edge as seen in the STD and error images. This is persistent across the three ice parameters. Still, as the STD and error are calculated based on an assumption of linear distance

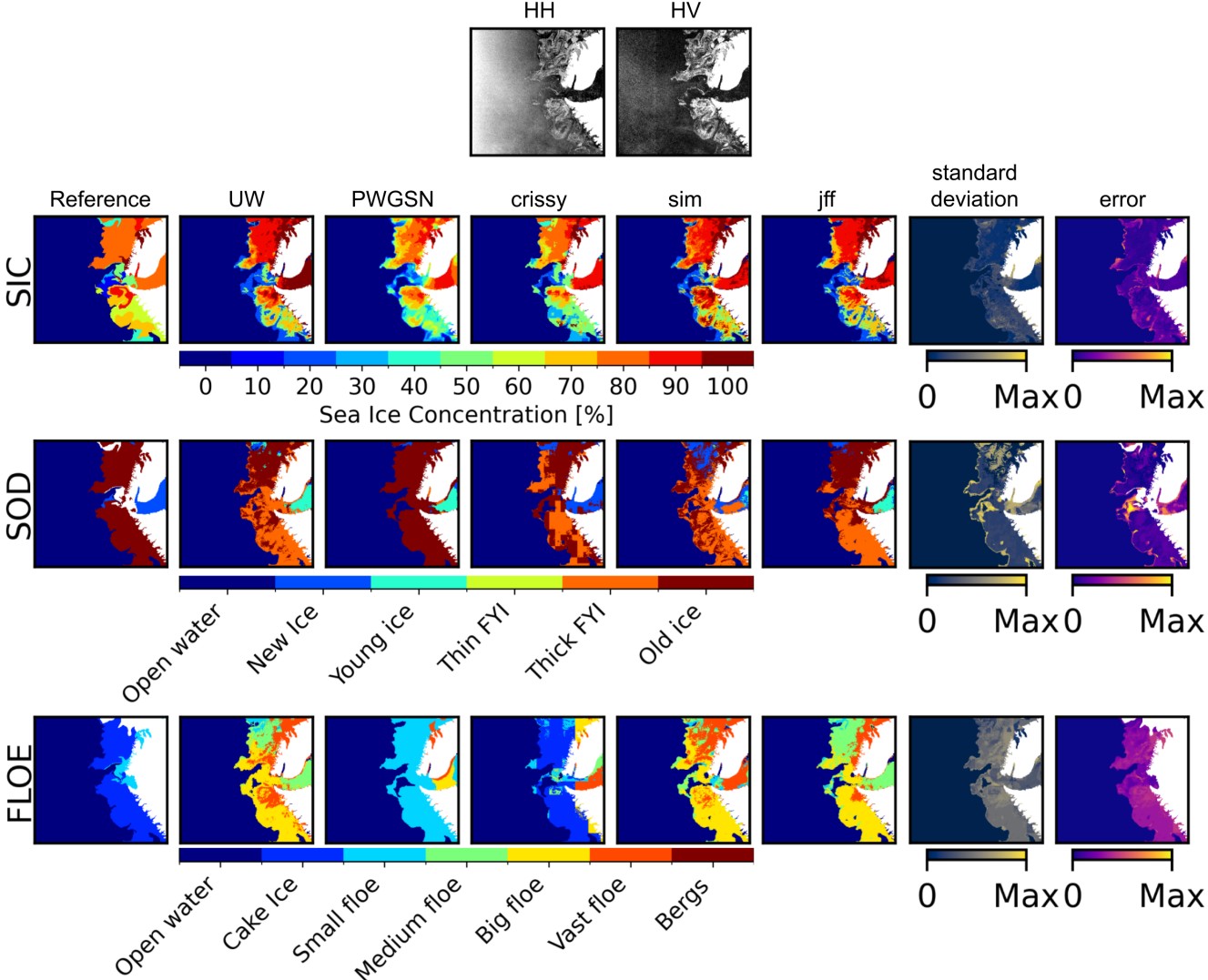

**Figure 11.** Scoresbysund, East Greenland. First row: SAR HH and HV images, acquired on October 10 2020. Reference ice chart labelled by Greenland Ice Service at DMI. Second row: SIC reference and top 5 solution SIC maps with standard deviation between solutions and accumulated map of error between solutions and the reference. Max indicates the maximum possible standard deviation of 4.9, 2.4 and 2.9 for SIC, SOD and FLOE, respectively, or max accumulated error assuming a linear distance between classes of 50, 25 and 30 for SIC, SOD and FLOE, respectively. The third row contains the SOD and the fourth the FLOE. White areas indicate a mask of either land, with no information or ice polygons without a dominant ice code.

between classes, the difference becomes most notable in the SOD as the difference between open water and *thick FYI* is large here. The FLOE parameter is also relatively consistent across the solutions. However, the *UW* team appear to have hit the location of the *vast floe* class given in the reference ice chart more accurately than the other teams.

The second scene example is illustrated in Fig. 11 with an ice chart labelled by the Greenland Ice Service at DMI, showcasing the large Scoresbysund fjord in Eastern Greenland. The scene was acquired in October 2020 and thus at the beginning of the cold period with *newly formed ice* in the fjord and *old ice* along the coast in *cake ice* form and *small floes*. This scene also contains many different SICs, varying SAR signatures, no strong wind patterns and dim ice signatures in the fjord. Again, the separation between open water and ice is strong, with all SIC solution maps capturing the complexity of the labels well with

low inter-solution STD and error. The solutions also, to a large extent, identified that the ice is old along the coast and *new / young* in the fjord. There is also a considerable SOD error in the centre of the image, which is caused by many of the solutions saying that open water is present here instead of *old ice*. As the SIC is low here, there is quite a bit of open water, implying that this error is not problematic but perhaps rather an expression of the ice charting methodology and the way polygons were drawn. This could be due to a tendency of the ice services to be conservative in their delineation of ice polygons (i.e.

the tendency in some cases to draw more ice than is present), a result of the coarser resolution of the ice charts (i.e. not all openings in the sea ice in the SAR imagery are resolved in the ice charts), or even a combination of both. The produced FLOE maps, however, have a large STD with only one solution correctly identifying the *cake ice* patterns along the coast. In contrast, three solutions label it as *big* or *vast floes*, which naturally gives rise to a significant error. Another notable feature is the model output submission from *crissy* that appears blocky. This is likely a result of outputting small sections of the map at a time and

stitching the image together, which typically limits the field of view of the model and its capability of producing continuously looking outputs.

**Table 4.** Average percentage sea ice parameter class accuracies, ± standard deviation and value range indicated by the minimum and maximum for SIC, SOD and FLOE for the top 5 participants. *Ice* implies ice pixels labelled as any true SIC above 0%. Intermediate SICs are compressed to one class, indicating the percentage of intermediate class predictions correctly labelled as a true intermediate class. Open water accuracies for SOD and FLOE are omitted for simplicity.

| SIC | Open water | Ice | Intermediate | 100% Ice | | |
|---|---|---|---|---|---|---|
| | 96.85% ±2.53% | 94.98% ±2.31% | 75.53% ±10.08% | 83.46% ±12.20% | | |
| | [92.01%, 98.57%] | [92.23%, 98.89%] | [66.92%, 93.53%] | [59.89%, 93.92%] | | |
| SOD | New Ice | Young ice | Thin FYI | Thick FYI | Old ice | |
| | 15.58% ±9.39% | 19.73% ±3.13% | 20.55% ±15.61% | 75.92% ±10.60% | 42.11% ±22.17% | |
| | [3.86%, 32.48%] | [15.85%, 24.62%] | [2.75%, 44.19%] | [58.16%, 89.11%] | [17.93%, 82.71%] | |
| FLOE | Cake Ice | Small | Medium | Big | Vast | Bergs |
| | 14.26% ±28.53% | 18.81% ±14.86% | 21.43% ±7.85% | 50.44% ±5.31% | 60.38% ±12.52% | 13.87% ±27.73% |
| | [0%, 71.32%] | [4.35%, 37.97%] | [12.61%, 35.64%] | [41.81%, 58.01%] | [36.52%, 71.21%] | [0%, 69.34%] |

    The average percentage class accuracies, ± standard deviation and value range indicated by the minimum and maximum for each sea ice parameter are presented in Tab. 4. For simplicity, the open water accuracy is only included under the SIC class performance, as this ice parameter contains the most pixels (due to some polygons not having dominant SOD or FLOE).

In addition, similar to Stokholm et al. (2023), SIC performance can be summarised using macro classes, open water, any ice

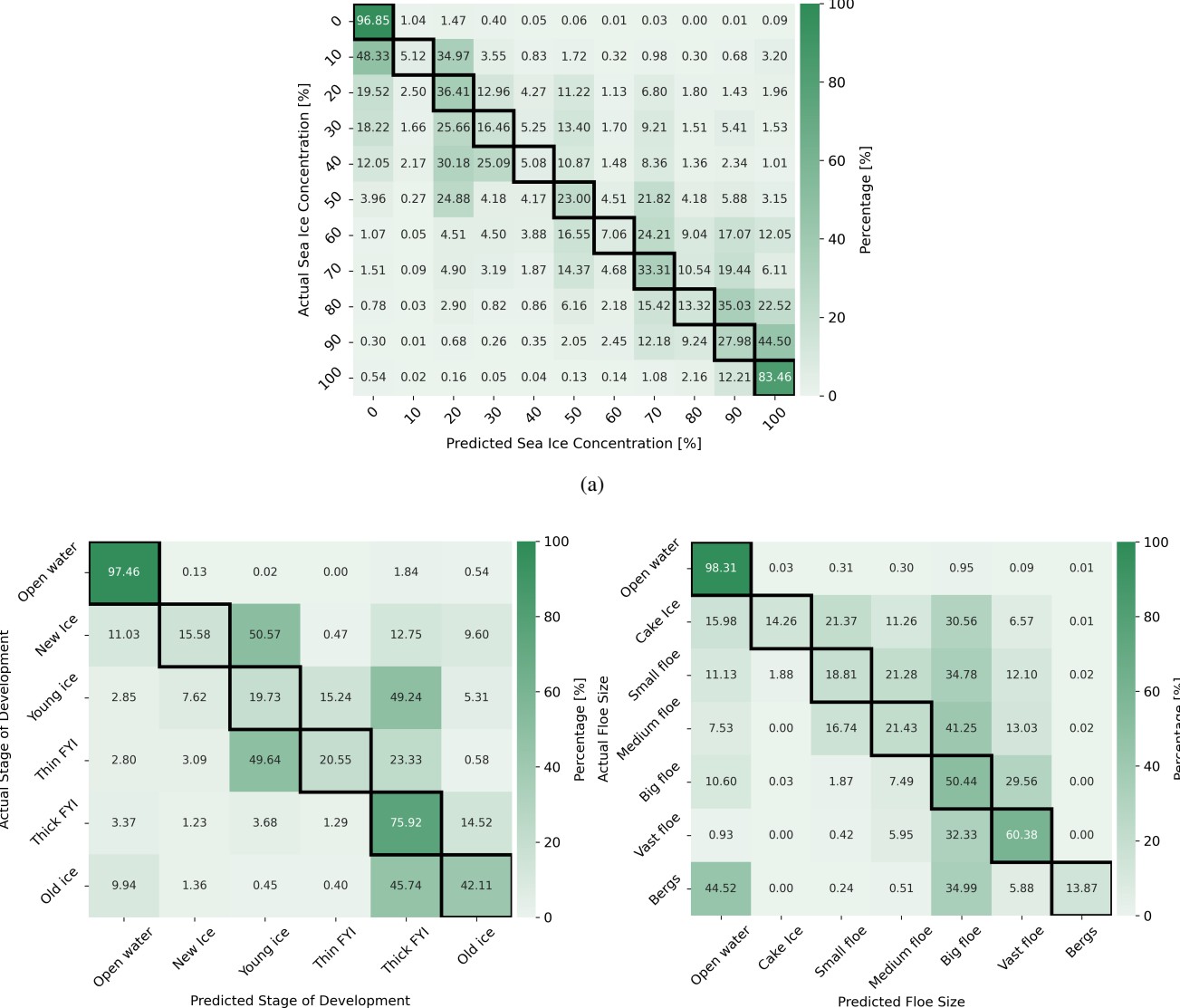

(a)

(b)                   (c)

**Figure 12.** Confusion matrices for **a** SIC, **b** SOD, **c** FLOE in percentages from 0-100%. Diagonal elements are highlighted with black borders and represent the accuracy for each class.

class ("*Ice*"), true intermediate pixels outputted by the model as any intermediate class and 100% sea ice. This is due to the relatively large uncertainties in the intermediate classes as highlighted in Karvonen et al. (2015) and Cheng et al. (2020), resulting in accuracy for individual SIC classes being uninformative. Here, the models' capabilities in separating water and ice are highlighted with a high open water accuracy of 98.65% and 94.98% of ice labelled as any SIC above 0% and associated low standard deviations. Further high accuracies are obtained in the intermediate and 100% ice categories of 75.53% and 83.46%,

respectively, though with higher standard deviations. For the SOD parameter, it is clear that *new* and *young ice*, as well as *thin FYI*, is challenging for the models, while the *thick FYI* has the highest score of 75.92% followed by *old ice* with an accuracy of 42.11% with a very large deviation in performance. Finally, the FLOE scores highlight difficulties with *cake ice*, and *small* and *medium floes*, while *big* and *vast floes* received higher accuracies of 50.44% and 60.38%, respectively. Finally, *bergs* were the most difficult, with an accuracy of merely 13.87%. *Cake ice* and *bergs* exhibit the largest performance difference with accuracies varying between 0 and 71.32% and 69.34%, respectively.

To expand on the class accuracies, confusion matrices for each sea ice parameter for the combined top-5 submissions are included in Fig. 12 with **a** SIC, **b** SOD, and **c** FLOE. The matrices show percentages of predicted classes in contrast to the actual classes. Each row sums to 100%, and the diagonal elements are demarcated with black borders and indicate the percentage of individual correctly labelled classes. In Fig. 12a, the predictions align well with the diagonal with notable exceptions of 10, 40, 60 and 80% where the submissions tend to produce fewer class outputs, indicating that the models prioritise the neighbouring classes. The deviation from the actual to the predicted intermediate SIC appears to be within $\pm 2$ classes, e.g. models predict 70% classes for actual classes of 50-90%. The submissions also have high accuracies for open water and 100% SIC. However, nearly half the actual 10% class and 90% class were predicted as open water and 100%, respectively.

For SOD in Fig. 12b, the same tendency with actual and predicted classes align with the matrix diagonal. It appears that the submissions classify *new ice* and *thin FYI* as *young ice* close to 50% of the time, whereas the actual *young ice* is predicted as *thick FYI* at the same rate. The models also appear to often label *old ice* as *thick FYI*.

In the FLOE confusion matrix in Fig. 12c, the predicted and actual classes align with the matrix diagonal. However, the majority of *cake ice*, *small floe* and *medium floe* are predicted as *big floe*, in addition to a notable portion of *vast floe* and *bergs*. Most actual *bergs* classes are predicted as open water. The test scene with *bergs* consists of SIC areas of less than 50% and the majority of 30% of less. In addition, if the areas mostly consist of small bergs, a lot of ocean will be present, which may confuse the models.

## 7   Discussion

Overall, the top-5 participants scored well on the selected metrics and showed strong separation between open water and sea ice. However, as indicated by Tab. 4 and Fig. 12, the models struggled to classify the SOD classes *New Ice*, *Young Ice* and *Thin FYI* correctly but tended to follow the matrix diagonal in the confusion matrices. Similarly, *Cake Ice*, *Small*, and *Medium* floes proved challenging for the top participants as well as the *Bergs* class. As the FLOE parameter score was substantially numerically lower than the SIC and SOD scores, additional research on improving it is warranted. It is plausible that participants gave less priority as the weight for this parameter was half that of SIC and SOD. This lower weight was assigned because the ice charting experts in the AI4Arctic external panel of experts deemed this parameter less critical for the ice service end users. The ice charts used as label data in the challenge are not produced with associated uncertainties for the SIC, SOD and FLOE information. Suppose the FLOE parameter is generally given less attention during the charting process. In that case, there might be a higher degree of uncertainty accompanying this parameter, so that the label quality could be lower.

## 7.1 Top-5 team approaches

**Table 5.** Summary of the 3 top 5 teams' approaches, including dataset version, the preprocessing steps taken, how the data was loaded, implementation details, such as model architecture, model optimisation approach, and finally the teams' technical background. *Learning Rate

| Team | Dataset | Preprocessing | Dataloader | Implementation | Experience |
|---|---|---|---|---|---|
| 1 - UW | RTT | downsample SAR, upsampling coarse resolution variables, latlon + time, data augmentation | get-started tools | U-Net architecture, cosine annealing LR* scheduling | sea ice + AI |
| 2 - PWGSN | RTT | data augmentation, upsampling coarse resolution variables | new sampling method | transfer learning and vision transformer | AI |
| 4 - sim | RTT | upsampling coarse resolution variables | get-started tools | U-Net architecture, constant LR* | space |

Tab. 5 summarises the main characteristics of the solutions presented by the three top-5 teams, including the version of the dataset used, the preprocessing steps taken, data-loading, implementation details, such as the model architecture and model optimisation, and finally the teams' technical experience. All three top-5 teams have used the RTT dataset. Two teams have used or modified the U-Net model provided, while two teams added data augmentation. All teams have applied the same approach to feeding the model different data types by upsampling coarse resolution variables with the get-started tools and ingesting it

with the SAR data. Two teams applied more advanced model optimisation strategies with cosine annealing learning rate and transfer learning with weights optimised on the ImageNet dataset (Russakovsky et al., 2015).

The three teams had different professional backgrounds, with sea ice domain experts, AI practitioners and space engineering knowledge, which is thought to have affected the variety of solutions presented and led to some interesting discussions. Domain knowledge allowed the *UW* team to tinker with the input and output, while AI expertise allowed for more advanced modelling

architectures in the *PWGSN* team. Fusing the two approaches could lead to further improvements, as suggested by the PWGSN team during the Winner's Event.

As the only team, *UW* applied additional preprocessing steps by downsampling the SAR data before ingesting it into the network. This increases the effective geographical field of view of the model, allowing it to see information further away when deciding the class for a particular pixel. This approach is contrary to the approaches presented in Heidler et al. (2021);

Stokholm et al. (2022), as these sought to achieve the same but by increasing the number of pixels in the model's receptive field instead of increasing the area each pixel covers. This increased pixel cover has a tradeoff with a loss of effective and detailed resolution but yields computational efficiency as fewer pixels are funnelled through the network. However, as the polygons in the ice charts are relatively coarse (except for the boundaries) compared to the SAR data, this loss of resolution does not appear to hamper the models from learning to replicate the human-produced SIC and SOD ice charts. However, we see that *UW* score

lower in FLOE, which could be because the delineations of the individual smaller ice floes are lost. It may also substantially reduce the training duration and memory requirements, allowing for quicker model iterations, which, in addition to *UW* being a large team, could have increased the iteration rate, triggering many submissions.

The *UW* team also provided additional information to the model regarding the geographical location by ingesting the latitude and longitude of the scene and the acquisition month. It is possible that knowing the location of the scene could be beneficial in determining the SIC and particularly SOD, as multiyear ice typically drifts South along the East coast of Greenland. In contrast, the West Coast of Greenland and the Baffin Bay area have less multiyear ice. The SAR scene acquisition time could also be beneficial for the model, enabling it to better capture the sea ice seasonal changes, especially for the SOD parameter. The combination of both the geographical location and the time of the year could be a powerful information combination for mapping SOD. This is supported in the ablation study in Chen et al. (2024a) with a 9.1% decrease in SOD performance when removing the information.

Among the top solutions, it is interesting that *UW* scored best on SIC and SOD — by a significant margin — but fifth on the FLOE parameter. This could reflect less effort towards this parameter, or perhaps the high amount of downsampling could blur the individual ice floe boundaries. If boundaries between smaller floes are difficult to distinguish, it could be difficult to differentiate *cake ice*, *small*, and *medium floes*, which could lead to lower performance in three of the FLOE classes. *Big* and *vast floes* do not appear to be problematic. Across the top teams, these observations align with the confusion matrix in Fig. 12c. This hypothesis could be further supported in the SIGRID-3 documentation with individual floe sizes of 30cm - 20m for cake ice, 20m - 100m for small floes and 100m - 500m for medium floes while considering that *UW* downsamples with a factor 10, giving a SAR pixel spacing of 800m. In addition, this may support using the native SAR pixel spacing of 40m instead of 80m, which is otherwise used in the challenge.

*UW* investigate the effect of SAR downsampling on the FLOE retrieval in Chen et al. (2024a), indicating a performance loss of 5.3% when their additional SAR downsampling is removed. At the same time, SIC and SOD lose 7.3% and 6.5%. However, given that the receptive field of the CNN model is not increased, the comparison is strictly between a lower and higher pixel coverage and, subsequently, an effective geographical field of view. In this context, the analysis supports an advantage in having a larger effective geographical field of view. Ultimately, this analysis cannot conclude whether the downsampling of FLOE delineations hampers performance compared to not downsampling as severely and instead relying on increasing the model's receptive field. Empirical evidence to support this analysis could be provided by comparing models with the same effective geographical field of view but achieved either through downsampling or by increasing the model's receptive field by expanding the number of pixels the model can access during individual pixel predictions.

An elaborate validation scheme was applied by the *UW* team, where scenes were selected to be approximate both geographically and temporally to the test set, enabled by the date given in the file names and the geographical coordinates provided in the files. This allowed the team to compare model outputs more frequently to scenes that may have some information leakage. While this validation selection is within the challenge's rules, it could have positively affected the team's scoring and final rank on the leaderboard. Ideally, the test scenes should have been selected sufficiently distant temporally to prevent any leakage from testing data to training or validation. This is a notable takeaway for organising similar competitions in the future.

The *PWGSN* team was the only one of the top-5 teams to apply vision transformers in their winning solution, which have been hailed as very potent for computer vision but require additional computational resources compared to the U-Net. The team used pre-configured weights (transfer learning), trained on ImageNet (Russakovsky et al., 2015), which contains large

quantities of RGB images. Here, the pre-trained weights included three-channel input for RGB. However, the competition data had 24 channels. Here, *PWGSN* choose to average the three-channel input weights and repeat them to match the 24 channels.

While this is practical, the weights trained on RGB images may not be suitable for the remote sensing and climate data, with particular emphasis on the AMSR2 and reanalysis data with less structured patterns than typical real-life images. However, this weight-averaging approach may have been the most feasible as the competition's data volume may not have been sufficient to train a vision transformer from scratch. In addition, *PWGSN* implemented an alternative dataloading scheme to speed up their training time and to mitigate class imbalance by sampling less frequently appearing classes.

Lastly, the *sim* team utilised the provided get-started tools and the provided U-Net model with tuned hyperparameters to perform well and brought the team to a top-5 ranking. Therefore, it can be noted that the supporting get-started tools provided to the participants worked well and allowed for competitive models. Multiple teams also investigated using the DeepLabV3 (Chen et al., 2017) architecture but did not achieve better results than the U-Net on this particular segmentation task.

### 7.2 The state of automatic sea ice mapping

Given the decisions on design choices regarding the dataset, metric selection, using sea ice charts, and selecting dominant ice classes in polygons using a threshold, the outcome of the challenge contains some bias despite attempts to minimise it. Nonetheless, given the large participation in the challenge, the top teams should represent the general field of automatic sea ice mapping and retrieval of the three sea ice parameters with overlapping shortcomings that can be extrapolated to the remaining community.

Regarding SIC, models appear capable of identifying open water and fully ice-covered areas, similar to earlier SIC works in Radhakrishnan et al. (2021); de Gelis et al. (2021); Tamber et al. (2022); Stokholm et al. (2023); Wulf et al. (2024). However, correctly assigning intermediate SIC appear to remain an obstacle, as presented in the SIC confusion matrix in Fig. 12a. Some classes appear underutilised, as the models do not appear to produce many 10, 40, 60 and 80% outputs often and instead prioritise neighbouring classes. The class imbalance could cause this, but it also appears similar to the results reported in

Kucik and Stokholm (2023) when utilising classification optimisation objectives. Given the $R^2$ metric, this is not penalised significantly but is apparent when inspecting the class accuracies. As not all classes are utilised by the model, it may be better to combine some neighbouring intermediate SIC classes to simplify the problem and mitigate class imbalance issues, as explored in Stokholm et al. (2023). However, the paper only assessed the impact of separating macro classes rather than combined $R^2$-score when combining classes.

Another notable element is the variation of the assigned intermediate SIC classes in Fig. 12a, appearing as actual classes distributed across $\pm 2$ predicted classes or 20% in many cases. In addition, lower SIC appears to be overestimated as higher SIC. Middle SIC, e.g. 50%, is widely spread, and higher SIC is skewed towards lower concentrations. These observations align with the results reported in Wulf et al. (2024).

The SOD classes *new ice* and *thin FYI* also appear difficult to predict correctly, given the SAR image, dominant polygon

ice classes and class distribution favouring *thick FYI* and *old ice*. Similarly, for FLOE, *cake ice small floe* and *medium floe*

predictions can be improved, which are, to a large extent, assigned as *big floe*. For FLOE, the class distribution favours *big* and *vast floe*.

Other factors could explain these shortcomings, as the electromagnetic SAR signatures could be ambiguous for these SOD and FLOE classes. There may also be underlying bias in the ice charting process, such as some classes are viewed as secondary and used occasionally instead of regularly. *Bergs* also appears difficult to map in the style of the sea ice charts. *Bergs* often cover few pixels and may often be in regions with low SIC, making the effective number of *berg* pixels small, and thus few examples for the model to train on.

## 7.3 Challenge considerations

To perform comparative studies of different SAR-based sea ice retrievals of SIC, SOD and FLOE to establish the state-of-the-art, there is a need for a standardised benchmarking dataset. Here, the challenge has provided an initial version of such a reference dataset with 20 scenes selected for testing the models. Naturally, evaluating the models more thoroughly with many more ice conditions, years, and geographical areas is desired and presents a key opportunity to further contribute to the automatic sea ice mapping community. Creating an online dashboard (Papers With Code, 2024) with sea ice retrieval test results akin to *ImageNet* (Deng et al., 2009) could drive further competition and innovation in this space.

One obstacle in the current evaluation of the SOD and FLOE is the occasional absence of a dominant ice class. Therefore, it may be helpful to evaluate on polygon level rather than pixel level, which could enable the use of the partial concentrations for the evaluation of SOD and FLOE. Despite some polygons' lack of dominant ice types, models can still produce segmentation results in these areas. Future models could be able to create maps with the individual partial polygon concentrations of the SOD and FLOE classes, effectively increasing the information resolution of the maps.

For evaluation, measuring SOD performance with macro classes similar to the SIC summary in Tab. 4 may be useful. Macro classes could combine *new* and *young ice*, as this would allow for, at least conceptually, macro categories with closely related ice types. Similarly, for FLOE, some combination of *cake*, *small*, and *medium floes* could be combined and *big* and *vast floe*.

Naturally, automatic sea ice mapping research could benefit from an increased dataset size with thousands of scenes and an accompanying larger testing dataset. This could help train better models and evaluate them more generally while minimising potential biasing in the choice of the current scenes. Work is underway to expand the current ASID Challenge dataset to provide the ASID-v3. However, with an increased number of scenes, the imbalance of intermediate SIC, younger SOD and floes of smaller size is expected to remain present. Therefore, work on incorporating better class balancing could improve performance. However, Kucik and Stokholm (2023); Stokholm et al. (2023) have experimented with class weight balancing when optimising SIC models, showing an increased performance on intermediate SIC but lower performance for open water and fully ice-covered areas. Another approach could be to sample classes with lower representation more often.

A limitation of the challenge and developing supervised deep learning models to map sea ice information based on sea ice chart reference data is the large polygons with low effective resolution and the SOD and FLOE information mixed within each polygon. An open question is whether mimicking these coarse and mixed maps is the future for sea ice mapping. With better and more reliable satellite communication, the polar regions have more transmission bandwidth to acquire sea ice information,

which could warrant moving towards more detailed sea ice maps. Nonetheless, using manually derived sea ice charts has fuelled supervised model developments with open-source and readily available expert-labelled information without needing large-scale annotation efforts of SAR imagery. However, the community may be ready to undertake such efforts.

## 8 Conclusions

This article presents the AI4EO AutoICE Challenge in full with the challenge setup, dataset description, and participation
statistics while briefly summarising 3 of the top-5 solutions before highlighting two test scenes with the top-5 participants' model output maps. Finally, a discussion that compares the different approaches is included with an assessment of the state of automatic sea ice mapping. The competition was won by the University of Waterloo team from the Department of System Engineering, followed by the teams *PWGSN*, *crissy*, *sim*, and finally *jff*. The challenge had 129 registered teams representing 179 users, with 494 submissions in total by 34 of the 129 teams comprising a participation rate of 26.4%.

Overall, the †AI4Arctic team is delighted with the extensive participation from across a broad and diverse international community ranging from sea ice and computer vision experts to students who have used the competition as part of their educational activities. The tools provided in the competition proved to be both competitive and useful to the participants, with the 3 top-5 teams described here using the ready-to-train dataset.

Through the competition, participants have proven that it is possible to perform multi-ice parameter retrieval with deep
learning models using professionally produced sea ice charts across multiple national ice services and national boundaries. Top solutions showed that the total sea ice concentration and stage of development were mapped the best, while the floe size was the most difficult. Furthermore, participants offered intriguing approaches and ideas that could help propel future research within automatic sea ice mapping. Particularly showing that higher rates of SAR data downsampling do not degrade model SIC and SOD performance when evaluated against ice charts but may not fully exploit the rich information in the SAR data.

Intermediate SIC remains difficult to assign correctly, with models overestimating low SIC, middle SIC having a wide spread while high SIC can be underestimated. Younger SOD classes and floe sizes that are less extensive are also difficult.

## 9 Future Work

The AI4Arctic Sea Ice Challenge Dataset (ASID Challenge) incorporated several additional data sources compared to the ASID-v2 dataset (Saldo et al., 2021), such as numerical weather prediction parameters. Mapping how influential each data
source is on both combined and individual ice parameter retrieval model performance is a natural next step in quantifying data fusion choices. In addition, all the top participants applied the provided data ingesting approach of upsampling coarse resolution data to the SAR pixel spacing, but this simple solution may be naive. Therefore, an investigation of alternative approaches could provide a more appropriate means of integrating the data sources.

The prospect of downsampling the SAR data yielding good results is promising and provides an avenue to reduce complexity
and hardware constraints in training models. However, more research into how downsampling affects the performance of the

three ice parameters would be beneficial, but investigating how to better utilise the rich information in the SAR data could also yield additional benefits.

Some of the issues with correctly assigning intermediate SIC, younger SOD classes and less extensive FLOE sizes could be addressed by having a larger dataset with more scenes. However, approaches to better optimise models on these classes should be investigated, such as weighted sampling. Furthermore, developing models capable of producing higher-resolution sea ice maps compared to the manually derived sea ice charts could yield further advantages of utilising an automatic approach.

Another option for future work is estimating uncertainties for all sea ice parameters. This, in connection with the mapped sea ice parameters, could be useful for both maritime navigation and as assimilation parameters into climate and weather models.

Finally, it will be possible for the AutoICE participants to continue their work when the next iteration of the ASID dataset, ASID-v3, is released. The new dataset will comprise 16 times as much data compared to the competition dataset, which will allow a much larger test dataset to be selected, much more data to train on and with the addition of ice charts from the Norwegian Ice Service (with SIC only), which further expands the geographical coverage.

*Code and data availability.* Data from the competition is available here: Buus-Hinkler et al. (2022a) (https://doi.org/10.11583/DTU.c.6244065. v2) and the code provided to participants are available here: Stokholm et al. (https://github.com/astokholm/AI4ArcticSeaIceChallenge)

*Video supplement.* A short video describing the AutoICE Competition: https://youtu.be/iuXIeLPyKfg

*Author contributions.* The AI4Arctic consortium consists of Andreas Stokholm, Jørgen Buus-Hinkler, Tore Wulf, Anton Korosov, Roberto Saldo, Leif Toudal Pedersen, David Arthurs, Rune Solberg, Nicolas Longépé and Matilde Brand Kreiner, and has been the main architects and composers of the AutoICE Challenge. The AI4EO consortium encompasses Ionut Dragan, Iacopo Modica, Juan Pedro, and Annekatrien Debin and was responsible for hosting the competition on the AI4EO platform. Jan N. van Rijn, Jens Jakobsen, Martin S. J. Roers, Nick Hughes, and Tom Zagon constituted the expert panel that supported the AI4Arctic consortium in designing the competition. Top participants were Xinwei Chen, Muhammed Patel, Fernando J. Pena Cantu, Javier Noa Turnes, Jinman Park, Linlin Xu, Andrea K. Scott, David A. Clausi, Yuan Fang, Mingzhe Jiang, Saeid Taleghanidoozdoozan, Neil C. Brubacher, Armina Soleymani, Zacharie Gousseau, Michał Smaczny, Patryk Kowalski, Jacek Komorowski and David Rijlaarsdam. The teams have each provided a section describing their solutions to the challenge.

The AI4Arctic consortium was managed by Rune Solberg and led by Matilde Brandt Kreiner. The ESA technical officer and project owner was Nicolas Longépé. The initial manuscript draft was prepared by Andreas Stokholm. The first review of the manuscript was conducted by Jørgen Buus-Hinkler, Matilde Brand Kreiner, Nicolas Longépé and Tore Wulf. All authors have reviewed and approved the initial manuscript submission.

*Competing interests.* The authors declare that there are no competing interests.

*Acknowledgements.* The authors would like to acknowledge all challenge participants for a constructive competition and ESA $\phi$-lab for funding the competition and providing the AI4EO platform to host it.

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
