# Peer review of "The AutoICE Challenge"

_EGUsphere, 2023_

## Author Response (AR1)

**DTU Space**
National Space Institute

**1st reply to reviewer comments for**
**"The AutoICE Challenge"**
**[EGUSPHERE-2023-2648]**

Dear Editor Juha Karvonen,

On behalf of my co-authors and myself, I would like to thank the three reviewers for their comments on our manuscript. The reviewers have made extensive suggestions for improving the explainability in key areas and scientific value. We have followed their recommendations to our best efforts in the vast majority of cases. Where it was not possible, we provided an extensive justification for why it was the case. An additional figure with confusion matrices has been included in the *Results (Section 6. Comparison of the top 5 submissions)* section, and the discussion has been overhauled and extended. In the following, we provide a point-by-point answer to all the issues raised by the reviewers. We have gathered all reviewers' points in black, our answers are highlighted in blue, the original text is in red, and the edited content is in green. In addition, we have uploaded an associated *difference* document highlighting the changes in the manuscript.

Best regards on behalf of the authors,
Andreas Stokholm

**Editor comments:**

Justification (visible to authors and reviewers only):

Some things to consider to be taken into account in a revised version:

1) Could the authors provide some text describing the algorithm of the other two top-5 teams (sort sections)?

We agree this would be an asset to the manuscript. Unfortunately, we have been unable to reach the remaining top 5 participants, and we do not have their models. Therefore, we cannot provide descriptions of their models. We have further clarified this at L246:

In the following subsections three of the top 5 teams - UW, PWGSN and sim teams - have contributed with short descriptions of their model solutions.

In the following subsections, three of the top 5 teams - UW, PWGSN and sim teams - have contributed short descriptions of their model solutions. The two remaining teams have not provided information about their personal models.

2) Are the related tool and algorithm source code publicly available? If so, please, provide references.

The open-source software provided to the participants is available in the *Code and data availability* statement. We have also included the reference in the manuscript body at L120 to make this more apparent.

3) The language may require checking at some points.

We have reviewed the manuscript and attempted to improve the language throughout.

4) One topic to mention as part of the future work could be including the related uncertainties. Uncertainty is important e.g. in data assimilation.

A paragraph regarding uncertainties for the sea ice parameters has been included in the future work section.

**Reviewer 1:**

General Comment:

- This article reads extremely well, with no issues with the English language use and with good clear content.

We appreciate the reviewer's general comment.

- However, there is just no scientific content or progress. The article would be a very nice summary for the competition web-pages, but it does not achieve any scientific goals.

We acknowledge that more scientific content could be added to the manuscript and have tried to address this, as highlighted in the following comment responses.

- There are some hints at scientific concepts, some summarising of the state-of-the-art, but they are only suggestions with no backup or supporting evidence.

To accommodate this comment, we have expanded the summary of related works in section 1.2 to include works on specific sea ice parameter retrieval literature entries while explaining how the field has evolved. In addition, we have included other scientific work in the discussion of the competition results.

- I conclude that this work could possibly be made valuable, but a total overhaul is needed to emphasise the scientific messages.

We acknowledge the need for added scientific content that is in line with the previous responses. To accommodate the comment, we have expanded our analysis of the top participants by adding confusion matrices for the three sea ice parameters. In addition, the discussion has been heavily expanded to provide more value to the reader. Please refer to the *difference* document attached, which highlights the changes made to the manuscript.

Specific Comments:

- Abstract: No scientific goals are mentioned. The competition may very well have encouraged novel development, but that is not explicitly explored. The organisers are in an unique position to inter-compare and condense the information contained in the choices of solutions and those that achieved best results. They do make some hints at categorising this, but it needs to go much further.

We acknowledge the need to explicitly mention the goal of the article in the abstract. We suggest revising the following in the abstract at L3:

The aim of the AutoICE Challenge was to encourage the creation of models capable of mapping sea ice automatically from spaceborne Synthetic Aperture Radar (SAR) imagery using deep

learning while inspiring participants to move towards multiple sea ice parameter model retrieval instead of the current focus on a single sea ice parameter, such as concentration.

The AutoICE Challenge investigates the possibility of creating deep learning models capable of mapping multi-sea ice parameters automatically from spaceborne Synthetic Aperture Radar (SAR) imagery and assesses the current state of the automatic sea ice mapping scientific field. This was achieved by providing the tools and encouraging participants to adopt the multi-sea ice parameter retrieval paradigm rather than the current focus on single sea ice parameters, such as concentration. The paper documents the efforts, analyses, compares and discusses the performance of the top five participants' submissions.

- I note that lines 94-95 and 101-102 do contain some aims to inter-compare and discuss, but these need to be made the formal goal and emphasis in the article. A "literature review" type paper is only worthwhile if the authors can add some value in the summary of the material and opinions on directions to take.

We have clarified the objective of the challenge and manuscript by adding more context in several locations.

L79:

The objective of the AutoICE challenge is to advance the state of the art for sea ice parameter retrieval from SAR data with an increased capacity to derive more robust and accurate automated sea ice maps.

The objective of the AutoICE challenge was to advance state-of-the-art sea ice parameter retrieval from SAR data with an increased capacity to derive more robust and accurate automated sea ice maps and show that models can retrieve multiple sea ice parameters. In parallel, this provides an opportunity to assess the current state of the scientific field.

L94:

This paper summarises and compares the top participants' results and discusses the outcome of the AutoICE Challenge.

This paper summarises the AutoICE challenge, the AI4Arctic Sea Ice Challenge Dataset, the tools provided to the participants, and the evaluation of submissions. In addition, the results of the top five participants are analysed and compared, and the outcome of the AutoICE Challenge and the state of the automatic sea ice mapping research field are discussed, highlighting avenues for future work.

- Section 7, the "Discussion" does attempt to suggest some good ideas and reasons for the results, but it is mostly speculation. Do the authors have any further evidence, or can they actually perform any experiments to demonstrate these ideas, perhaps with the assistance of the competition teams who are listed as co-authors? The value would be hugely more meaningful with further logical or experimental support.

To address this comment, we have made large changes to the discussion, both to incorporate the literature more while assessing the model performances more thoroughly and to evaluate

the state of the sea ice mapping research field. Due to the extent of the changes, we refer to the *difference* document for specific changes.

- Lines 160-167: When introducing the sea ice charts, it may be worth emphasising the very coarsely drawn polygons, as this is a well known aspect of the manually drawn charts. The limitations of using known coarse and mixed regions should probably have further discussion, since it is being discussed in the community. Do we want to mimic these coarse charts, or create something more detailed. The egg-codes do account for partial mixtures, but machine-learning training does not usually manage this. Further discussion of this in the motivation and discussions would be valuable.

We have expanded on the description of the manually derived sea ice charts in the manuscript at L160:

Sea ice charts describe the local ice condition at the capture time, based on professional interpretations of SAR images and represented distinctly as polygons of relatively homogeneous areas of sea ice, steered by the common guidelines outlined in the SIGRID-3 standard but still subject to individual interpretation.

Sea ice charts describe the local ice conditions at the time of acquisition of the input satellite imagery, based on professional interpretations of SAR images and represented distinctly as polygons of relatively homogeneous areas of sea ice, steered by the common guidelines outlined in the SIGRID-3 standard but still subject to individual interpretation. There is a natural limitation to how many details and polygons the ice analysts can manually draw within the chosen scale and coverage of the ice map. At the same time, there is a focus on safety and not delaying the information to the users more than necessary. Therefore, the polygons' boundaries are very accurately drawn but cover large areas with a subsequent low effective resolution. Understandably, manual production cannot relay ice information with a level of detail that matches the high-resolution and multidimensional electromagnetic SAR textures.

To address the mixed sea ice SOD and FLOE classes within polygons, we have added the following at L179-183:

In addition, as the SOD and FLOE are given as partial SOD or FLOE concentrations, there may be multiple categories of SOD or FLOE for each ice polygon. To select the SOD or FLOE category while avoiding ambiguity, the SOD or FLOE category must be dominant. Here, we defined a SOD or FLOE category as dominant if said category has a partial concentration of at least 65\%. Therefore, there are numerous polygons where a total SIC exists, but the polygon does not have an associated SOD and/or FLOE.

In addition, as the SOD and FLOE are given as partial SOD or FLOE concentrations, there may be multiple categories of SOD or FLOE mixed within each ice polygon without the exact location provided. To select the SOD or FLOE class while minimising ambiguity, the SOD or FLOE class must be dominant. Here, we defined a SOD or FLOE class as dominant if the associated partial concentration is at least 65\%. Therefore, there are numerous polygons where a total SIC

exists, but the polygon does not have an associated SOD and/or FLOE. Despite this effort, multiple classes may still be mixed in each polygon.

In addition, we have added a paragraph at the end of the discussion regarding the use of the coarse sea ice charts.

A limitation of the challenge and the associated work with developing supervised deep learning models to map sea ice information based on sea ice chart reference data is the large polygons with low effective resolution and the SOD and FLOE information mixed within each polygon. An open question is whether mimicking these coarse and mixed maps is the future for sea ice mapping. With better and more reliable satellite communication, the polar regions have more transmission bandwidth to acquire sea ice information, which could warrant moving towards more detailed sea ice maps. Nonetheless, using manually derived sea ice charts has fuelled supervised model developments with open-source and readily available expert-labelled information without needing large-scale annotation efforts of SAR imagery. However, the community may be ready to undertake such efforts.

Technical Comments:

- Line 159: This line appears to start a list, but the list is lost in the following sub-headings, with no clear termination. I suggest two options: Firstly, since the sub-sections are quite long, I suggest that this leading sentence actually lists them by name and completes the list with a comment like "explained below"; Secondly, add sub-section numbering or indentation to make it clear which items are in the list, and when the list stops and does not continue into the next sub-section.

We understand that this may appear confusing, so we have taken the first suggestion:

Each of the scenes contains:

Each scene contains sea ice chart reference data, SAR images, passive microwave radiometry measurements, and numerical weather prediction parameters. The following subsections describe these data sources and a prepared and ready-to-train dataset.

- Line 169: I believe that the description of the colours should be "lighter colour" and not "brighter colour". Brighter usually implies stronger/richer/saturated colour. While "lighter" can mean with more white added (which is used in the figure), and the converse "darker" would be towards black.

We have revised the paragraph accordingly.

- The sentence on lines 178-179 is very unclear. Please explain more clearly the relationship between the values, sub-categories of ice, and the polygon.

We have attempted to make this clearer in response to the comment made concerning L160-167.

- Lines 259-261: These sentences should be either better joined, or thematically separated. At the moment, the first sentence says that it incorporates both spatial and temporal information by adding lat-lon coordinates only. The temporal is in the separate sentence. Please get make the logic correct.

We have tried to clarify by joining the two sentences in the following way:

Additionally, to incorporate spatial and temporal information, we interpolate the latitude and longitude coordinates of the Sentinel-1 SAR geographic grid points to match the size of the input SAR image. The acquisition month of each SAR scene represents the time information for each pixel.

Additionally, to incorporate spatial and temporal information, we interpolate the latitude and longitude coordinates of the Sentinel-1 SAR geographic grid points to match the size of the input SAR image and add the acquisition month of each SAR scene to each pixel to represent the time information.

- Lines 292-294: You are adding weights p and q. Do these act like prior probabilities, abundances? Are they likely to be image, region or season dependent?

p and q represent probabilities for sampling these types of patches but without taking the image, region or season into account. We propose adding this additional clarifying information:

During an epoch, patches were randomly selected from scenes to approximately satisfy the predefined class distribution, implying that the share of ice-only patches in an epoch should be close to some value, \textit{p}, and the share of water-ice edge patches should be close to another value, \textit{q}.

During an epoch, patches were randomly selected from scenes to approximately satisfy the predefined class distribution, implying that the share of ice-only patches in an epoch should be close to some value, \textit{p}, and the share of water-ice edge patches should be close to another value, \textit{q}, but without taking image, region or season into account.

- Line 296: Probably need a "The" in front of "Following data augmentations", and subsequent capitalisation change.

This has been adjusted in the manuscript.

- Lines 299-300: There is something grammatically wrong with "This approach enabled us to make SIC, SOD and FLOE maps simultaneous predictions." Are you meaning that you make the predicted three image "maps" simultaneously?

We propose the following modification:

A modified semantic segmentation model was used with an adjusted number of output heads. This approach enabled us to make SIC, SOD and FLOE maps simultaneous predictions.

A modified semantic segmentation model was used with an adjusted number of output heads. This approach enabled us to make SIC, SOD and FLOE maps simultaneously.

- Line 303: Something is grammatically wrong with "the checkpoint used loss function". Is the word "used" necessary? Or, if this a specific phrase, then add hyphens or quote or emphasis to indicate this.

We recognise that this can be a confusing statement. Thus we have tried to rephrase the sentence:

All of the models in the ensemble shared the same architecture but differed in the checkpoint used loss function, augmentations and the imposed data distribution.

All of the models in the ensemble shared the same architecture but differed in the saved model parameter checkpoint, loss function, augmentations and the imposed data distribution.

- Line 317: It seems that CE is listed twice, or are some list items compound (with two pieces)? Perhaps it needs further commas to separate the compound elements, or say how many "best" results you are listing, or clarify the meaning some other way.

We agree that this can be confusing. We suggest:

The best results were obtained with CE, WCE and CE and dice loss mixture with the ratio of $\frac{0.7}{0.3}$, respectively.

The best results were obtained with CE, WCE and a mixture of CE and dice loss with the ratio of $\frac{0.7}{0.3}$, respectively.

- Line 445: Suggest "...was the only one of the top-5 teams...".

This has been rephrased in the manuscript.

**Reviewer 2:**

The authors present a good summary of the "AutoICE Challenge" competition. All the relevant points about the competition, including the objective, dataset, good solutions, and future work, are clearly and well written. Spaceborne SAR has demonstrated great potential for operational monitoring of sea ice in polar regions. The AutoICE challenge competition is an interesting initiative. The presented top solutions use various deep learning methods, from U-net to DeepLabV3+, as well as computer vision technique. They all yield good results on sea ice concentration, sea ice types and ice floes. This suggests that combination of SAR observations and machine learning is a promising solution for retrieving various sea ice parameters in much higher spatial resolution compared with other conventional remote sensing data.

We appreciate the reviewer's general comments on the manuscript.

However, the current manuscript is a project summary or report, instead of scientific paper, which is not suitable for the journal. If the authors would like to summarize state of the art of SAR observations of sea ice based on deep learning methods, together with the AutoICE challenge results, which might be more interesting to the scientific community. In fact, even before the AutoICE challenge starting, there are quite a few studies have applied various deep learning methods to derive sea ice information by spaceborne SAR.

We have sought to extend the state-of-the-art analysis of automatic sea ice mapping and its history. In addition, a deeper analysis of the top submissions has been carried out by adding confusion matrices for the three sea ice parameters. Furthermore, we have expanded the discussion of the challenge results while adding other literature to the discussion. Due to the extent of the changes, we refer to the *difference* document for specific changes.

**Reviewer 3:**

General comment:

This article gives a comprehensive overview of the AutoICE challenge, including the data set, evaluation metrics, and summaries of the top contributions. It is very well written and easy to follow, there are almost no language or formatting issues. While I think that the manuscript is worthwhile reading for anyone working in the field of sea ice remote sensing and ice charting, it is in its current form unfortunately not a scientific. The authors do not formulate a research question and do not provide a detailed evaluation of any scientific novelty.

The authors appreciate the general comment regarding the quality of the written manuscript. We have attempted to clarify the purpose of the challenge and the manuscript as emphasised in our response to reviewer 1.

While some novelty and scientific description (although more technical than scientific) is given in the summary of the top submissions to the competition in Section 5, the authors mention that there will be individual contributions of these solutions to a Cryosphere special issue on the AutoICE challenge. Simply repeating a summary of those contributions in this manuscript does not present any additional value. I note that the data set itself is of course a valuable scientific contribution - however publication of the data set does not warrant publication of this manuscript in its current form.

While we have encouraged the participants to submit their own manuscripts to further describe and investigate their solutions, only the winners of the competition have submitted an article for publication.

We have attempted to expand the scientific aspects of the manuscript by adding a broader overview of the literature and a deeper analysis of the results. We have also extended the discussion and added comparisons to other literature. Due to the extent of the changes, we refer to the *difference* document for specific changes.

The authors hint at several topics that could be investigated in detail (such as for example the influence of spatial resolution on the results for floe size distribution or the inherent uncertainty in the ice charts that are used for training and evaluation). I think that, with further work, the results and contributions of the competition could lead to valuable scientific contributions on these (or similar) research questions.

We hope that our modifications to the manuscript will assist in addressing this with our expansion of the discussion that focuses on the influence of the spatial resolution.

Technical/specific comments:

[Figure]

I only have very few technical or language related comments, most of which have already been mentioned in RC1. Since I think that the manuscript needs significant changes before it can be published, I will not list wording details here, but only give two general technical comments for any revised version of this manuscript:

- Please be consistent in use (or no use) of Oxford comma.
We have reviewed the manuscript and attempted to be consistent.

- I find that the different viewing geometries of the data presented in Fig 2,3,4 make it difficult to compare the content. I'd suggest using the same geometry and projection.
While we understand that the viewing geometries can be confusing, we prefer retaining the current geometries with emphasis on the original SAR viewing angles, which, in this case, are not projected. In addition, this mirrors the dataset and better encapsulates the data description, as the dataset users will be faced with the same viewings.

**1st reply to editor comments for**
**"The AutoICE Challenge"**
**[EGUSPHERE-2023-2648]**

Dear Editor Juha Karvonen,

On behalf of my co-author and myself, I would like to thank you for considering our manuscript for publication in The Cryosphere. We have addressed your raised points to the best of our abilities. In the following, we provide a point-by-point answer to all the points raised. We have tried to gather all editor points in black, our answers are highlighted in blue. We have updated the *difference* document to reflect the new changes based on our response to the editor comments.

Best regards on behalf of the authors,
Andreas Stokholm

**Editor comments:**

Dear authors of the manuscript EGUSPHERE-2023-2648,

This manuscript is aimed to be an overview of a TC special issue on the AutoICE challenge and the revised version includes the most important related information on dataset, methods and results. there does not seem to be many submitted manuscripts to this special issue (yet). If the authors have up-to-date information on the coming contents of the special issue it could be included in the manuscript also.

We are not in possession of up-to-date information on potential upcoming manuscripts to this special issue but hope more participants have intention to submit entries on their work.

I have a few concerns related to the manuscript:

1) Section 5 "Top five submission solutions" still only contains subsections of the Rand 1,2 and 4 solutions. Is it really so that there is no information at all of the missing two solutions? And how can this be possible? If possible include even short subsection on the missing two approaches including any available information. Such challenge should require at least some kind of description of the methodology from all the participants. Otherwise it would be impossible to verify the credibility of the results either. Or is this a copyright issue? This should be taken into account in possible similar challenges in the future. I hope it would be possible to say at least something also on the missing two top 5 solutions. it would be consistent to have one subsection for each top 5 submission.

We agree that it would be most consistent to have a subsection for each top 5 submission. However, as much as we would like to describe the remaining two top five solutions, we are not in possession of any information regarding their approaches. Submitting solutions to the

competition only required uploading a file with the proposed solution in a specific format. As the participants could upload any number of solutions, we feared that requiring participants to include detailed descriptions of their approaches for each submission could limit the number of participants or solutions in the competition. Instead, in order to receive a winning prize, we made it mandatory for participants to partake and describe their solution during a *Winner's Event*. Subsequently, the two missing teams chose not to partake in the event and neither received their prize. Another approach we could have taken would have been to disqualify the two teams and instead select the two proceeding teams. However, that seemed inappropriate at the time, skewing the results. It is also worth noting that the participants own any intellectual property resulting from their contributions to the competition, which makes it difficult to force participants to do something unwillingly. For future competition, it may be useful to rethink this approach, as you suggest, to ensure open and transparent results.

2) Section 6:"Comparison of top 5 submissions": would it also be possible to give some measures of deviation e.g. in Table 4. E.g. ranges and/or standard deviations could be included. This is a great suggestion. In Table 4, we have added the standard deviation as +- and the range indicated by the minimum and maximum obtained values. This is described in the associated text and included in relevant areas in Section 6.

Also check the references and update according to the most recent information. For example, the reference by Chen et al. (2023) seems to be available on TC and has a DOI also, in the updated version it should become Chen et al. (2024).
Well spotted. The reference Chen et al. has been updated.

---

## Author Response (AR2)

**2nd reply to editor comments for**
**"The AutoICE Challenge"**
**[EGUSPHERE-2023-2648]**

Dear Editor Juha Karvonen,

On behalf of my co-author and myself, I would like to thank you again for considering our manuscript for publication in The Cryosphere. We have addressed your raised points to the best of our abilities. In the following, we provide a point-by-point answer to all the points raised. We have tried to gather all editor points in black. Our answers are highlighted in blue, the previous text in red, and our modifications in green. We have updated the *track changes* document to reflect the new changes based on our response to your comments.

Best regards on behalf of the authors,
Andreas Stokholm

**Editor comments:**

Dear authors of the manuscript egusphere-2023-2648,

The manuscript has improved from the previous version and many of the reviewer comments and my comments have been taken into account.
We are pleased that you found the revised manuscript improved.

The important value of the article is that a good reference data set has been created and the challenge participants have provided a good reference performance measures that can be used testing new (continuously developing) machine learning methods for sea ice parameter estimation.
This could be emphasized even more in the introduction and/or discussion/conclusion sections. For future developments it is very useful to have a reference data set and also some reference estimation results. The provided data set and the challenge have provided these.
We agree that the reference dataset is a valuable contribution to the automatic sea ice mapping community. We suggest adding the following modifications to the manuscript at L87:
Furthermore, the challenge has provided a common reference dataset that can be used as a benchmark for comparisons of future model developments.
and L95:
In addition, the challenge provides a common reference benchmark to support further comparisons in future model developments within the community.
In the discussion at L547:
In the challenge, 20 scenes were selected for testing the models. Naturally, evaluating the

models more thoroughly with many more ice conditions, years, and geographical areas than in the test set is necessary. To perform comparative studies of different SAR-based sea ice retrievals of SIC, SOD and FLOE to establish the state-of-the-art, there is a need for a standardized benchmarking dataset.

To perform comparative studies of different SAR-based sea ice retrievals of SIC, SOD and FLOE to establish the state-of-the-art, there is a need for a standardised benchmarking dataset. Here, the challenge has provided an initial version of such a reference dataset with 20 scenes selected for testing the models. Naturally, evaluating the models more thoroughly with many more ice conditions, years, and geographical areas is desired and presents a key opportunity to further contribute to the automatic sea ice mapping community. Creating an online dashboard [ImageNet 2024] with sea ice retrieval test results akin to *ImageNet* [Deng 2009] could drive further competition and innovation in this space.

I noticed that at least the following reference is missing in the reference list:
X. Chen, M. Patel, L. Xu, Y. Chen, K. A. Scott and D. A. Clausi,
Weakly Supervised Learning for Pixel-Level Sea Ice Concentration Extraction Using AI4Arctic Sea Ice Challenge Dataset, in IEEE Geoscience and Remote Sensing Letters, vol. 21, pp. 1-5, 2024, Art no. 1500205, doi: 10.1109/LGRS.2023.3338061.

This reference can be used to justify the use of ice charts as training data for higher-resolution estimation. There are also other possibilities, e.g. automatically manipulating the ice chart training data within the ice chart segments based on SAR data and the segment-wise information and to create a high-resolution target classification this way before applying the actual machine learning algorithm. In any case, utilizing ice chart information in training to automatically provide higher-resolution estimates is an important future research topic.
We agree that this article is worth mentioning in the related works section at L82:
Furthermore, the authors in [Chen 2024] have shown that detailed SIC maps can be obtained by training models with sea ice charts.
The aspect of future work has been highlighted in future works at L610
Furthermore, developing models capable of producing higher-resolution sea ice maps compared to the manually derived sea ice charts could yield further advantages of utilising an automatic approach.

It is unpleasant that not much information on some of the challenge participants could not be included in the manuscript. I hope for the possible future challenges related to this valuable data set the participants will at least be required to provide a thorough description of the algorithm and results in a form of a document. Additionally, source code would of course be preferable. Without a proper documentation the methods and the results can not be properly evaluated.

We agree that it would have been preferable to include information on the remaining two top-5 teams. Ultimately, this needs to be considered in the design phase of new challenges. Perhaps inspiration for accomplishing this can be gained from *ImageNet*.

After these minor revisions, I think the manuscript can be published as the overview article of the AutoICe challenge TC special issue.

We are pleased with your assessment and hope the modifications presented here are satisfactory.